# Multi-Scale Adaptive Network
# for Single Image Denoising

**Yuanbiao Gou**[1], **Peng Hu**[1], **Jiancheng Lv**[1], **Joey Tianyi Zhou**[2], **Xi Peng**[1]*

[1]College of Computer Science, Sichuan University, China.
[2]Institute of High Performance Computing, A*STAR, Singapore.
`{gouyuanbiao, penghu.ml, joey.tianyi.zhou, pengx.gm}@gmail.com`
`lvjiancheng@scu.edu.cn`

## Abstract

Multi-scale architectures have shown effectiveness in a variety of tasks thanks to appealing cross-scale complementarity. However, existing architectures treat different scale features equally without considering the scale-specific characteristics, *i.e.*, the within-scale characteristics are ignored in the architecture design. In this paper, we reveal this missing piece for multi-scale architecture design and accordingly propose a novel Multi-Scale Adaptive Network (MSANet) for single image denoising. Specifically, MSANet simultaneously embraces the within-scale characteristics and the cross-scale complementarity thanks to three novel neural blocks, *i.e.*, adaptive feature block (AFeB), adaptive multi-scale block (AMB), and adaptive fusion block (AFuB). In brief, AFeB is designed to adaptively preserve image details and filter noises, which is highly expected for the features with mixed details and noises. AMB could enlarge the receptive field and aggregate the multi-scale information, which meets the need of contextually informative features. AFuB devotes to adaptively sampling and transferring the features from one scale to another scale, which fuses the multi-scale features with varying characteristics from coarse to fine. Extensive experiments on both three real and six synthetic noisy image datasets show the superiority of MSANet compared with 12 methods. The code could be accessed from https://github.com/XLearning-SCU/2022-NeurIPS-MSANet.

## 1 Introduction

In the real world, images are often contaminated by various signal-dependent or -independent noises during the image acquisition process. As a result, the imaging quality will deteriorate, thus hindering people and computers from receiving image information. To solve this problem, image denoising, as an essential step for image perception, has been extensively studied in the past decades [10, 19, 27, 50].

In the early, filtering-based methods remove the image noise by manually designing low-pass filters, *e.g.*, median filtering [8], bilateral filtering [36], and wiener filtering [7]. Afterward, model-based methods remove the image noise by optimizing a problem of maximum a posteriori [13, 40, 43]. For instance, ITS [40] proposed an intrinsic tensor sparsity regularization on the non-local similar image patches by assuming they could be sparsely represented. Looking from the other side, both the filtering- and model-based methods are based on image priors from the statistics of natural images, and thus could be referred to as prior-based methods. Although remarkable performance has been achieved, an unpleasant denoising result will be obtained once the priors are inconsistent with the real data distribution. To avoid prior engineering, learning-based methods adopt a data-driven fashion to remove noise by learning the mapping from the noisy image to the corresponding clean image.

---

*Corresponding author

36th Conference on Neural Information Processing Systems (NeurIPS 2022).

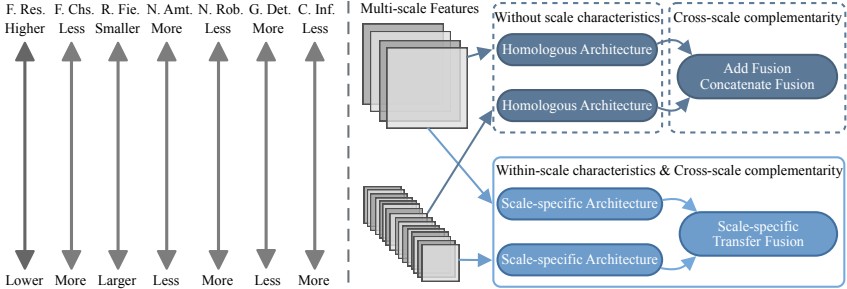

Figure 1: The motivations of MSANet. Left: the features at different resolutions show varying characteristics. In the figure, "F. Res." and "F. Chs." denote feature resolutions and channels, respectively; "R. Fie." denotes receptive field; "N. Amt." and "N. Rob." denote noise amount and robustness, respectively; "G. Det." and "C. Inf." denote geometric details and contextual information, respectively. Right: the major difference of MSANet with existing multi-scale architectures, *i.e.*, different scale features show varying characteristics and should be processed by scale-specific structures rather than homologous architectures.

With the popularity of deep neural networks, various network architectures have been designed and achieved state-of-the-art performance [6, 26, 32, 51]. For example, DnCNN [50] introduced residual learning and batch normalization to implement a denoising convolution network. SwinIR [19] introduced the shifted windowing scheme to implement a restoration Transformer network.

Among the network design paradigms, multi-scale architectures play a significant role in performance improvement thanks to multi-scale features. However, existing studies design their network architectures by only considering the cross-scale complementarity while ignoring the within-scale characteristics (see Fig. 1). To be exact, the shallow and high-resolution features, which lack awareness of contextual information, are sensitive to inputs and contain a lot of noises. Nevertheless, they contain abundant image geometric information, such as edges and textures. In consequence, they are critical for the recovery of fine-grained image details, and their network structures should take the advantages and make up for the disadvantages. The deep and low-resolution features are competitive in noise-robustness and contextual information, which are critical for the recovery of coarse-grained image context, but the too-low resolution will destroy the image structure. Hence, their network structures should also make full consideration of their characteristics during the architecture design. To summarize, as different scale features show varying characteristics, it is deserved to deal with them via scale-specific structures to fully adapt to their characteristics.

Based on the motivation, we propose a novel Multi-Scale Adaptive Network (MSANet) for single image denoising, which simultaneously incorporates the within-scale characteristics and the cross-scale complementarity into architecture design by overcoming the following three difficulties, *i.e.*, i) how to adaptively sample image details and filter noises; ii) how to adaptively extract contextually informative features without changing feature resolutions; iii) how to adaptively fuse the multi-scale features with varying characteristics. Accordingly, three neural blocks, *i.e.*, adaptive feature block (AFeB), adaptive multi-scale block (AMB) and adaptive fusion block (AFuB) are proposed. In brief, AFeB handles the features with mixed details and noises through adaptively sampling and weighting, and thus the image details are preserved while filtering noises. AMB extracts the contextual information through dilated convolutions and adaptive aggregation, and thus contextually informative features are obtained while keeping the resolution unchanged. AFuB adaptively samples and transfers the features from one scale to another scale, and the multi-scale features with varying characteristics are fused from coarse to fine.

To summarize, the contributions are as follows:

- We propose a novel neural network for single image denoising, termed as MSANet. The major difference with existing methods is that MSANet simultaneously considers and incorporates the within-scale characteristics and the cross-scale complementarity into multi-scale architecture design, which is a missing piece before and the first revelation so far.

- To exploit the within-scale characteristics and achieve the cross-scale complementarity, we design three neural blocks, *i.e.*, AFeB, AMB, and AFuB, which are used to implement our

idea, *i.e.*, building scale-specific subnetworks corresponding to different scale features by considering their characteristics.

- Extensive experiments are conducted on three real and six synthetic noisy image datasets to show the effectiveness of MSANet, and the significance of the within-scale characteristics.

## 2 Related Work

In this section, we briefly introduce existing single image denoising methods and multi-scale architectures, and discuss the major differences between MSANet and them.

### 2.1 Single Image Denoising

In general, most existing single image denoising approaches could be categorized as prior- and learning-based methods. Prior-based methods are based on some priors from natural images, such as local smoothing [41], self-similarity [36, 10, 13], and sparsity [43, 40]. For instance, BM3D [10] eliminates noisy pixels by transforming the 3D stack of non-local similar patches and employing a shrinkage function to obtain sparse coefficients. WNNM [13] introduced a low-rank weight coefficient based on the nuclear norm minimization and exploited the non-local similar image patches to remove noise. Different from prior-based methods that heavily rely on handcrafted priors, learning-based methods learn the mapping from the noisy image to the clean image in an end-to-end manner. In recent, a large number of methods have been proposed and achieved state-of-the-art performance [6, 14, 32, 38, 47]. For example, MemNet [35] proposed a persistent memory network to fuse both short- and long-term memories for capturing different levels of information. FFDNet [51] enhanced the denoising network for non-uniform noise by using the noise level map. Non-local attention [37] was designed to exploit the image self-similarity, and NLRN [20] incorporated it into a recurrent neural network. SADNet [6] proposed residual spatial-adaptive block and multi-scale context block to constitute a denoising network. DeamNet [32] introduced an adaptive consistency prior and designed an interpretable deep denoising network. With the rise of vision Transformers, Uformer [38] proposed a U-shape Transformer based on the locally-enhanced window Transformer block and multi-scale restoration modulator. Restormer [47] introduced the channel-based self-attention and gating mechanism to implement an efficient Transformer.

Different from the aforementioned methods, MSANet proposed three neural blocks, *i.e.*, AFeB, AMB, and AFuB, by simultaneously considering the within-scale characteristics and the cross-scale complementarity, to constitute the scale-specific subnetworks corresponding to different scale features for adapting their varying characteristics.

### 2.2 Multi-Scale Architecture

Multi-scale architectures have played a significant role in many fields of computer vision [5, 17, 21, 48], thanks to multi-scale features and their cross-scale complementarity. The straightforward way for multi-scale architecture is to separately feed multi-/single-resolution images/features into single/multiple subnetworks, and then fuse the outputs as a result [18, 25, 39, 44, 45]. For example, HRNet [34] proposed a multi-scale network by gradually adding high-to-low resolution subnetworks and repeating multi-scale fusions for human pose estimation. CLEARER [12] proposed a multi-scale neural architecture search to automatically determine where to fuse multi-scale features for image restoration. GDN [22] exploited the multi-scale information using a grid-like network and employed an attention mechanism to improve the performance. DID-MDN [49] proposed a multi-stream densely connected network to efficiently leverage features of different scales for image deraining. MSCNN [33] consists of a coarse-scale network and a fine-scale network to learn a transmission map for image dehazing. PANet [25] proposed a pyramid attention network for image restoration by capturing long-range feature correspondences from a multi-scale feature pyramid. In addition, [6, 11, 38, 47] employed encoder-decoder architecture to combine the high-to-low with low-to-high resolution features through skip-connections.

Although the aforementioned studies and our work share similarities in multi-scale architecture, they are remarkably different. The existing methods only consider the cross-scale complementarity and use homologous architectures for different scale features. In contrast, our work additionally considers

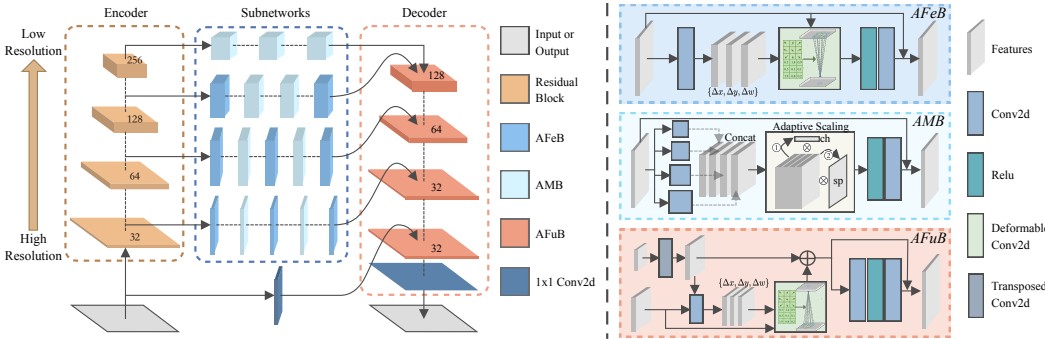

Figure 2: The framework of MSANet. It employs an asymmetric encoder-decoder architecture with multiple subnetworks to capture and fuse the scale-specific features. In addition, three neural blocks are designed to exploit the within-scale characteristics and achieve the cross-scale complementarity of multi-scale features. Note that we take four scales of features as a showcase and for experimental evaluations, more scales are allowed in practice.

the within-scale characteristics and designs scale-specific structures to embrace both, which is the missing piece and the first revelation for multi-scale architecture design.

## 3 Proposed Method

In this section, we propose a novel single image denoising network, *i.e.*, MSANet, which simultaneously embraces the within-scale characteristics and the cross-scale complementarity of multi-scale features through the three neural blocks. In the following, we will first introduce the architecture principles of MSANet and then elaborate on the three neural blocks.

### 3.1 Architecture Principles

As illustrated in Fig. 2, MSANet employs an asymmetric encoder-subnetworks-decoder architecture. Specifically, the encoder adopts four residual blocks [15] to extract features of four scales. The first residual block aims to extract the initial features without changing the resolution. The other residual blocks respectively decrease the resolutions to half while increasing the channels to double. With the multi-scale features extracted by the encoder, the subnetworks aim to exploit their within-scale characteristics via AFeB and AMB blocks. As the characteristics of multi-scale features gradually change from high- to low-resolution, we take the two bottom and two top subnetworks in Fig. 2 as the high- and low-resolution branches, respectively.

The two bottom subnetworks need to handle the high-resolution features which usually are characterized by a mixture of details and noises. It is highly expected to remove the noises without losing the fine-grained image details, and thus AFeB is designed for adaptively preserving the indispensable details and filtering unpleasant noises. Meanwhile, another characteristic of high-resolution features is the limited contextual information. The corresponding subnetworks should also enrich the contextual information while keeping the feature resolution unchanged in order not to lose details. As a result, AMB is designed for adaptively extracting contextually informative features. As AFeB and AMB could mutually boost each other, *i.e.*, low noise features from AFeB allow AMB to better capture the image contexts, while contextually informative features from AMB help AFeB to better distinguish the noises and details, we alternately stack AFeB and AMB to build the subnetworks. In addition, as the higher-resolution features usually have fewer channels (*i.e.*, fewer parameters), a deeper subnetwork is allowed to further alleviate their disadvantages, such as massive noises and limited contexts.

Different from the bottom subnetworks, the two top subnetworks aim to exploit the within-scale characteristics of the low-resolution features, which are characterized by less noise and more contextual information. As the too-low resolution will destroy the image structure, the structure consistency cannot be guaranteed during the image recovery. Hence, we employ AMB to capture more image contexts for enriching the contextual information without reducing the feature resolution. In addition,

as the low-resolution features are more robust to noise and have more channels, the top subnetworks mainly consist of AMB and are with shallower depth for efficiency. Meanwhile, except for the lowest resolution, the first and the last blocks in the top subnetworks are AFeB for adaptively controlling the input and output features.

To achieve the cross-scale complementarity, the decoder consists of four AFuB blocks followed by a convolution layer to output the recovered image. More specifically, the first three AFuB blocks increase the resolutions to double and decrease the channels to half, while adaptively sampling and transferring the fine-grained detail features to the coarse-grained contextual features. The last AFuB block keeps the resolution and channel unchanged, and directly samples and transfers the image details from the noisy input, which is an effective alternative to the global skip-connection, for which could avoid introducing noises from the noisy input.

For a given noisy input $x$, MSANet employs the encoder to extract the features of different scales and then feeds the features into different subnetworks to learn the scale-specific features. After that, the decoder fuses the multi-scale features with varying characteristics from coarse to fine to obtain the recovered image $\hat{y} = f(x)$, where $f(\cdot)$ denotes MSANet. To optimize MSANet, we employ the following objective function,

$$\mathcal{L} = \|y - \hat{y}\|_p^p \tag{1}$$

where $y$ is the ground truth of $x$, and $p = \{1, 2\}$.

### 3.2 Adaptive Neural Blocks

In this section, we elaborate on the proposed three neural blocks which are designed to learn better multi-scale features for single image denoising.

**Adaptive Feature Block (AFeB).** To preserve the indispensable details and filter unpleasant noises, the block is expected to adaptively sample and weight the input features $f_i$ based on themselves, *i.e.*,

$$\{\Delta x, \Delta y, \Delta w\}_{(x,y)} = F(f_i), \tag{2}$$

where $F(\cdot)$ is used to compute the offset $(\Delta x, \Delta y)$ w.r.t. the positions $(x, y)$, as well as the corresponding weight $\Delta w$. Then, the output features $f_{i+1}$ could be aggregated by

$$f_{i+1}(x, y) = \sum_{j=1}^{k} w_j * f_i(x + \Delta x_j, y + \Delta y_j) * \Delta w_j, \tag{3}$$

where $k$ is the number of samples, and $w$ denotes the learnable weights. In this way, AFeB could learn the sampling locations to indicate where are important for restoration, while assigning different weights to show how important the locations are. As a result, AFeB preserves the indispensable details and filters unpleasant noises for better restoration. However, it is prohibitive to traverse all positions for sampling and weighting w.r.t. each position in the input features. Hence, for the convenience of implementation, AFeB employs the modulated deformable convolution [55] to implement the aggregation operation in Eq.(3). In detail, AFeB sets the sample number to the kernel size and the learnable weights as the convolutional weights. Although this setting would decrease the number of samples and the flexibility of weights, it is efficient in the calculation of high-resolution features, and the limitations could be alleviated by stacking AFeB blocks. In summary, AFeB consists of a convolution layer $F(\cdot)$, a modulated deformable convolution, a LeakyReLU layer, a convolution layer and a skip-connection, *i.e.*,

$$f_{out} = f_i + F_{conv}(F_{relu}(f_{i+1})). \tag{4}$$

**Adaptive Multi-scale Block (AMB).** Contextually informative features are highly expected for both the high- and low-resolution branches. For the high-resolution features, reducing the resolution leads to the loss of image details. For low-resolution features, a too-low resolution destroys the structure consistency of recovered images. Therefore, to capture more contextual information without changing the resolution, we propose AMB by using several convolutions with different dilation rates. Convolution with a large dilation rate could provide a large receptive field, and multiple convolutions with different dilation rates could smoothly capture multi-scale information. To reduce the cost caused by the multiple convolutions, AMB compresses the channels of each convolution so that the concatenated channels of all convolutions are equal to the output channels, *i.e.*,

$$f_{i+1} = Concat(\{F_k^d(f_i)|d, k \in \mathbb{N}^+\}), \tag{5}$$

where $f_i$ and $f_{i+1}$ are input and output features, and $F_k^d$ is the $k$-th convolution with the dilation rate $d$. As the concatenation assigns different scale features into different channels, the distinctive importance of multi-scale features is not considered. To address this issue, AMB adaptively scales different channels and features, *i.e.*,

$$
\begin{aligned}
ch &= 2 * F_{sig}(F_{fc}(avg\_pool(f_{i+1}))), \\
f_{i+2} &= ch * f_{i+1}, \\
sp &= 2 * F_{sig}(F_{conv}(mean(f_{i+2}))), \\
f_{i+3} &= sp * f_{i+2},
\end{aligned}
\tag{6}
$$

where $avg\_pool$ is an adaptive average pooling on space domain, $mean$ denotes a mean operation along the channels, $F_{fc}$ is a linear layer, $F_{conv}$ is a convolution layer, $F_{sig}$ is a sigmoid layer, and $2*$ is used to control the amplification ($> 1$) or suppression ($< 1$). With the $f_{i+3}$, AMB passes it through a LeakyReLU layer, a convolution layer and a skip-connection, *i.e.*,

$$
f_{out} = f_i + F_{conv}(F_{relu}(f_{i+3})).
\tag{7}
$$

**Adaptive Fusion Block (AFuB).** As the high-resolution features contain a lot of disordered fine-grained image details, and the low-resolution features contain abundant coarse-grained image contextual information, it is desirable to transfer the fine-grained image details into the coarse-grained image context. To this end, AFuB first upsamples the coarse-grained features to the resolution of the fine-grained features via

$$
f_{coarse} = F_{TConv}(f_{coarse}^{low}),
\tag{8}
$$

where $F_{TConv}$ is a transpose convolution. Then, to address the issue of the disordered details, AFuB adaptively samples and weights the fine-grained features by using coarse-grained features to provide contextual information and fine-grained features to provide details information, *i.e.*,

$$
\{\Delta x, \Delta y, \Delta w\}_{(x,y)} = F(f_{coarse}, f_{fine}),
\tag{9}
$$

where $F(\cdot, \cdot)$ is used to compute the offset $(\Delta x, \Delta y)$ w.r.t. the positions $(x, y)$, as well as the corresponding weight $\Delta w$. After that, AFuB transfers the fine-grained details to the coarse-grained context via

$$
f_{coarse}^{fine} = f_{coarse} + \sum_{j=1}^{k} w_j * f_{fine}(x + \Delta x_j, y + \Delta y_j) * \Delta w_j,
\tag{10}
$$

where $k$ is the number of fine-grained detail features, $w$ is the learnable weights. Similar to AFeB, AFuB employs a convolution layer as the function $F$ and a modulated deformable convolution to achieve the aggregation in Eq.(10). Finally, AFuB uses a convolution layer, a LeakyReLU layer, a convolution layer, and a skip-connection to further refine features, *i.e.*,

$$
f_{out} = f_{coarse}^{fine} + F_{conv}(F_{relu}(F_{conv}(f_{coarse}^{fine}))).
\tag{11}
$$

## 4 Experiments

In this section, we first introduce the experimental settings, and then show the quantitative and qualitative results on nine datasets. Finally, we perform analysis experiments including ablation study and feature-based visualization. Due to space limitations, we present more experimental details and results in the supplementary material.

### 4.1 Experimental settings

We evaluate the MSANet on both real and synthetic noisy datasets. For the evaluations on real noise, we employ the SIDD [1], RENOIR [3], Poly [42] datasets for training, and use SIDD Validation, Nam [28] and DnD [30] datasets for testing. For synthetic noise, we train MSANet on DIV2K [2] dataset, which contains 800 images of 2K resolution, by adding Additive White Gaussian Noise (AWGN) with the noise levels of 30, 50, and 70. We use color McMaster [52] (CMcMaster), color Kodak24 (CKodak24), color BSD68 [24] (CBSD68) for testing color image denoising, and grayscale McMaster (GMcMaster), grayscale Kodak24 (GKodak24), grayscale BSD68 (GBSD68) for testing grayscale image denoising.

We implement MSANet in Pytorch [29] and carry out all experiments on Ubuntu 20.04 with GeForce RTX 3090 GPUs. In our implementations, we use four scales of features with channels of 32, 64, 128 and 256. Moreover, we train MSANet 100 epochs via $L_1$ loss for real noise and 300 epochs via $L_2$ loss for synthetic noise. Both real and synthetic noise training are with the batch size of 16 and the patch size of 128. To optimize MSANet, the Adam [16] optimizer is used, and the learning rate is initially set to 1e-4 and decays to zero via the cosine annealing strategy [23]. During the training, we randomly crop, flip and rotate the patches for data augmentation. In the testing, we employ PSNR and SSIM to evaluate the performance.

## 4.2 Comparisons on Real Noise Images

Table 1: Quantitative results on SIDD sRGB validation dataset.

| Method | CDnCNN-B | CBM3D | CBDNet | PD | RIDNet | SADNet | DeamNet | MSANet |
|--------|----------|-------|--------|------|--------|--------|---------|--------|
| PSNR | 26.21 | 30.88 | 33.07 | 33.96 | 38.71 | 39.46 | 39.47 | **39.56** |
| SSIM | - | - | 0.8324 | 0.8195 | 0.9052 | 0.9103 | 0.9105 | **0.9118** |

Table 2: Quantitative results on Nam dataset with JPEG compression.

| Method | CDnCNN-B | CBM3D | CBDNet | PD | RIDNet | SADNet | DeamNet | MSANet |
|--------|----------|-------|--------|------|--------|--------|---------|--------|
| PSNR | 37.49 | 39.84 | 41.31 | 41.09 | 41.04 | 42.92 | 42.03 | **43.52** |
| SSIM | 0.9272 | 0.9657 | 0.9784 | 0.9780 | 0.9814 | 0.9839 | 0.9790 | **0.9863** |

Table 3: Quantitative results on DnD sRGB dataset.

| Method | CDnCNN-B | CBM3D | FFDNet+ | CBDNet | N3Net | PR | RIDNet | SADNet | DeamNet | MSANet |
|--------|----------|-------|---------|--------|-------|------|--------|--------|---------|--------|
| PSNR | 32.43 | 34.51 | 37.61 | 38.06 | 38.32 | 39.00 | 39.26 | 39.59 | 39.63 | **39.65** |
| SSIM | 0.7900 | 0.8507 | 0.9415 | 0.9421 | 0.9384 | 0.9542 | 0.9528 | 0.9523 | 0.9531 | **0.9553** |

Real noise image denoising is challenging due to the real noise being usually signal-dependent and spatial-variant hinges on the in-camera pipeline. Therefore, we carry out denoising experiments on three real noise image datasets, *i.e.*, SIDD Validation, Nam, and DnD. In brief, the validation dataset of SIDD contains 1,280 $256 \times 256$ noisy-clean image pairs captured by the smartphone. Nam includes 15 large image pairs with JPEG compression, and we evaluate MSANet on the selected 25 $512 \times 512$ patches by following CBDNet [14]. DnD contains 50 pairs of real noisy-clean images captured by cameras, and 1,000 $512 \times 512$ patches are extracted for testing. Due to the ground truths of the patches are not publicly available, we obtain the PSNR and SSIM results via the online submission system [30]. Besides, since JPEG compression makes the noise more stubborn on the Nam, we first train our model on the combination of SIDD and RENOIR for the evaluations on SIDD Validation and DND, and then fine-tune the trained model on the Poly for the evaluations on Nam.

We compare MSANet with 10 denoising methods that are comparable in model complexity, *i.e.*, CDnCNN-B, CBM3D [9], FFDNet+, CBDNet, N3Net [31], PD [54], PR [46], RIDNet [4], SADNet and DeamNet. In experiments, we use the corresponding pretrained models provided by their authors, and refer to their results reported in the online submission system and papers.

Table 1 shows the quantitative results on SIDD validation dataset. In brief, MSANet achieves the highest PSNR and SSIM values compared to other methods, *e.g.*, 0.85dB, 0.1dB, 0.09dB gains in PSNR, and 0.0066, 0.0015, 0.0013 gains in SSIM over the RIDNet, SADNet, and DeamNet, respectively. For visual comparisons in Fig. 3, CBDNet and PD result in residual noises and pseudo artifacts, RIDNet, SADNet and DeamNet severely destroy the textures and obtain over-smoothed results. In contrast, our method MSANet recovers textures and structures more subtly and obtains clearer restoration. Some areas are highlighted by color rectangles and zooming-in is recommended for better visualization.

The quantitative results on Nam dataset are shown in Table 2, which demonstrates that our method achieves significant improvements over the other methods. Specifically, MSANet outperforms RIDNet with 2.48dB (0.0049), SADNet with 0.6dB (0.0024), DeamNet with 1.49dB (0.0073) in PSNR (SSIM) values. For the visual comparisons shown in Fig. 4, our method obtains the best result for details recovery and noises removal, which is closer to the ground truth than other results.

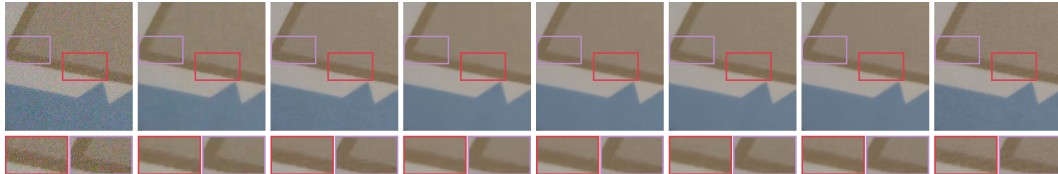

Figure 3: Qualitative results on real noise image from SIDD validation dataset. From left to right, we show the real noise image, the results of CBDNet, PD, RIDNet, SADNet, DeamNet, MSANet, and the ground truth.

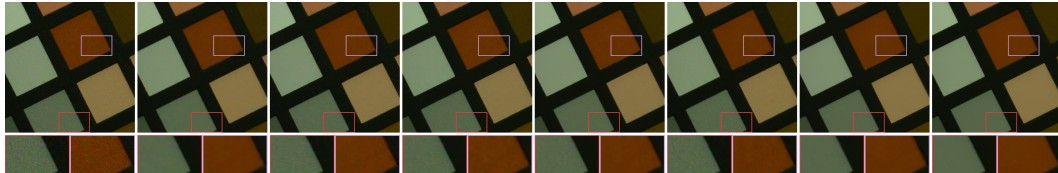

Figure 4: Qualitative results on real noise image from Nam dataset. From left to right, we show the real noise image, the results of CBDNet, PD, RIDNet, SADNet, DeamNet, MSANet, and the ground truth.

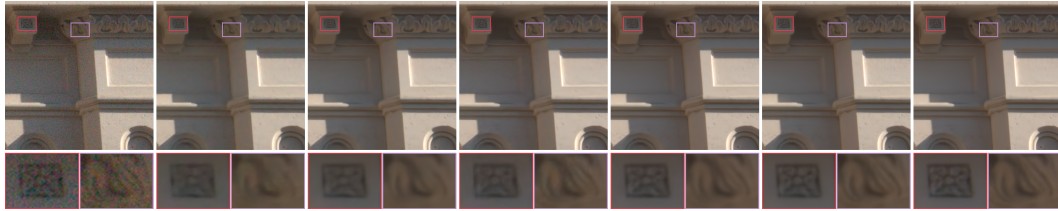

Figure 5: Qualitative results on real noise image from DnD dataset. From left to right, we show the real noise image, the results of CBDNet, RIDNet, PD, SADNet, DeamNet, and MSANet.

Table 3 reports the quantitative results on DnD dataset, which are obtained from the DnD benchmark website. From the table, one could observe that MSANet outperforms all methods both in PSNR and SSIM values. Moreover, we further perform a qualitative comparison on the DnD dataset. As shown in Fig. 5, the other methods achieve blurred results wherein many image details are corroded by noises, while our method can effectively remove the noises and obtain clearer details.

## 4.3 Comparisons on Synthetic Noise Images

Table 4: Quantitative results on synthetic color noise image datasets.

| Dataset | $\sigma$ | CBM3D | DnCNN | FFDNet | CLEARER | SADNet | RNAN | DeamNet | MSANet (Ours) |
|---|---|---|---|---|---|---|---|---|---|
| CMcMaster | 30 | 29.58/0.8107 | 29.64/0.8098 | 30.05/0.8221 | 30.83/0.8522 | 31.96/0.8857 | 32.01/0.8848 | 32.00/0.8862 | **32.07/0.8876** |
| | 50 | 25.92/0.7153 | 25.99/0.7147 | 26.23/0.7244 | 28.92/0.8143 | 29.72/0.8374 | 29.69/0.8333 | 29.78/0.8393 | **29.82/0.8403** |
| | 70 | 23.12/0.6398 | 23.03/0.6297 | 23.19/0.6406 | 26.96/0.7504 | 28.25/0.7988 | 28.14/0.7918 | 28.27/0.8000 | **28.35/0.8028** |
| CKodak24 | 30 | 30.33/0.8417 | 30.77/0.8548 | 30.62/0.8542 | 31.17/0.8590 | 31.72/0.8730 | 31.72/0.8716 | 31.76/0.8736 | **31.78/0.8744** |
| | 50 | 27.28/0.7572 | 27.63/0.7718 | 27.54/0.7687 | 28.94/0.7977 | 29.49/0.8149 | 29.43/0.8102 | 29.53/0.8155 | **29.57/0.8169** |
| | 70 | 24.84/0.6890 | 24.90/0.6912 | 24.88/0.6890 | 27.59/0.7503 | 28.10/0.7715 | 27.99/0.7635 | 28.14/0.7721 | **28.17/0.7731** |
| CBSD68 | 30 | 29.22/0.8378 | 29.72/0.8556 | 29.51/0.8526 | 30.35/0.8665 | 30.63/0.8749 | 30.61/0.8733 | 30.65/0.8749 | **30.67/0.8758** |
| | 50 | 26.06/0.7378 | 26.48/0.7600 | 26.38/0.7550 | 28.01/0.7996 | 28.31/0.8089 | 28.25/0.8050 | 28.34/0.8093 | **28.36/0.8107** |
| | 70 | 23.70/0.6548 | 23.86/0.6626 | 23.80/0.6584 | 26.58/0.7433 | 26.91/0.7577 | 26.81/0.7511 | 26.92/0.7574 | **26.96/0.7591** |

We carry out experiments on three color and three grayscale noisy image datasets. Specifically, the datasets are obtained by adding AWGN with the levels of 30, 50, and 70 to the color and grayscale version of BSD68, Kodak24 and McMaster, respectively. In brief, BSD68 contains 68 images which are frequently used for measuring image denoising performance, Kodak24 contains 24 images captured by film cameras, and McMaster contains 18 images with statistics closer to natural images.

For comparisons, we choose seven representative denoising methods, *i.e.*, BM3D [10], DnCNN [50], FFDNet [51], CLEARER [12], RNAN [53], SADNet [6] and DeamNet [32]. We call the python

Table 5: Quantitative results on synthetic grayscale noise image datasets.

| Dataset | $\sigma$ | BM3D | DnCNN | FFDNet | CLEARER | SADNet | RNAN | DeamNet | MSANet (Ours) |
|---|---|---|---|---|---|---|---|---|---|
| GMcMaster | 30 | 29.45/0.8151 | 29.80/0.8119 | 29.89/0.8292 | 30.29/0.8491 | 30.92/0.8649 | 30.92/0.8629 | 30.94/0.8656 | **30.96/0.8661** |
| | 50 | 26.23/0.7218 | 26.24/0.7281 | 26.46/0.7319 | 28.31/0.7945 | 28.61/0.8052 | 28.57/0.8014 | 28.65/0.8070 | **28.68/0.8072** |
| | 70 | 23.78/0.6517 | 23.63/0.6682 | 23.64/0.6466 | 26.83/0.7427 | 27.18/0.7606 | 27.06/0.7526 | 27.20/0.7616 | **27.22/0.7620** |
| GKodak24 | 30 | 28.71/0.7854 | 29.21/0.7946 | 29.15/0.8077 | 29.49/0.8132 | 29.87/0.8238 | 29.89/0.8208 | 29.90/0.8241 | **29.91/0.8248** |
| | 50 | 26.22/0.6996 | 26.52/0.7190 | 26.52/0.7177 | 27.27/0.7412 | 27.77/0.7559 | 27.73/0.7494 | 27.79/**0.7567** | **27.81**/0.7564 |
| | 70 | 24.37/0.6393 | 24.31/0.6647 | 24.28/0.6419 | 26.12/0.6931 | 26.51/0.7090 | 26.42/0.6989 | 26.53/**0.7107** | **26.54**/0.7091 |
| GBSD68 | 30 | 27.43/0.7721 | 27.96/0.7762 | 27.89/0.7982 | 28.27/0.8112 | 28.58/0.8165 | 28.59/0.8140 | 28.59/0.8165 | **28.61/0.8174** |
| | 50 | 24.90/0.6715 | 25.19/0.6826 | 25.19/0.6909 | 26.09/0.7295 | 26.50/0.7382 | 26.46/0.7333 | 26.50/0.7392 | **26.51/0.7393** |
| | 70 | 23.07/0.5985 | 23.04/0.6107 | 22.98/0.5942 | 25.03/0.6734 | 25.23/0.6828 | 25.15/0.6736 | 25.23/**0.6831** | **25.25**/0.6826 |

library for the evaluation of BM3D, and the pretrained models, provided by authors or retrained by us, for the evaluations of the other methods.

Table 4 reports the quantitative results on color image denoising, which shows that MSANet achieves the highest PSNR and SSIM values. Taking the noise level of 70 as an example, our method can achieve PSNR gains about $0.03 \sim 0.27$dB, and SSIM gains about $0.0010 \sim 0.0111$ over the state-of-the-art methods, *i.e.*, SADNet, RNAN, and DeamNet. Table 5 shows the quantitative results on grayscale image denoising. From the table, one could observe that MSANet achieves the highest PSNR values, and outperforms the other methods about $0.01 \sim 3.59$dB w.r.t. PSNR.

### 4.4 Analysis Experiments

The ablation study is conducted to demonstrate the significance of utilizing the within-scale characteristics (WSC) and the cross-scale complementarity (CSC). As shown in Table 6, "ED" and "ResB" are with homologous architectures, which use uniform Identity Mapping and Residual Block to process different scale features, respectively. As using AFeB and AMB alone cannot exploit WSC well, "AFeB"/"AMB" and "AFeB+AFuB"/"AMB+AFuB" only slightly improve the performance over "ResB" and "AFuB", respectively. When using AFeB and AMB together, "AFeB+AMB" and MSANet (*i.e.*, "AFeB+AMB+AFuB") significantly improve the performance over "ResB" and "AFuB", verifying our claim on the role of AFeB+AMB w.r.t. WSC. Furthermore, thanks to CSC through AFuB, "AFuB" and MSANet (*i.e.*, "AFeB+AMB+AFuB") are significantly better than "ResB" and "AFeB+AMB", respectively. In summary, the ablation study not only demonstrates the significance of utilizing WSC and CSC, but also shows the effectiveness of our proposed solution.

Table 6: Ablation study on CMcMaster with the noise level of 30. "ED" denotes the encoder-decoder architecture with skip connections. "ResB" denotes to substitute the blocks in MSANet with the residual blocks. "AFeB", "AMB", "AFuB", "AFeB+AMB", "AFeB+AFuB" and "AMB+AFuB" denote to use the corresponding blocks on the basis of "ResB".

| Ablations | ED | ResB | AFeB | AMB | AFuB | AFeB+AMB | AFeB+AFuB | AMB+AFuB | MSANet |
|---|---|---|---|---|---|---|---|---|---|
| PSNR | 31.70 | 31.93 | 31.94 | 31.94 | 32.01 | 31.98 | 32.04 | 32.03 | 32.07 |
| SSIM | 0.8801 | 0.8851 | 0.8854 | 0.8851 | 0.8864 | 0.8860 | 0.8869 | 0.8866 | 0.8876 |

To further demonstrate the significance of the within-scale characteristics, and the effectiveness of our solution in exploiting it, we show the qualitative comparisons on the intermediate multi-scale features before (*i.e.*, without) and after (*i.e.*, with) our subnetworks in MSANet. From the Fig. 6, one could observe that our subnetworks could well exploit the within-scale characteristics, *i.e.*, preserving the indispensable details while filtering the unpleasant noises for high-resolution features (*i.e.*, 2nd and 3rd columns), and capturing rich contextual information for low-resolution features (*i.e.*, 4th and 5th columns). Besides, the different scale features show varying characteristics and significant cross-scale complementarity, which would be further exploited by our AFuB.

## 5 Conclusions

In this paper, we proposed MSANet with three neural blocks, *i.e.*, AFeB, AMB, and AFuB, for single image denoising. Different from existing multi-scale architectures, MSANet considers not only the cross-scale complementarity but also the within-scale characteristics, thus boosting the recovery performance as verified in experiments. As this work could be regarded as finding the missing piece

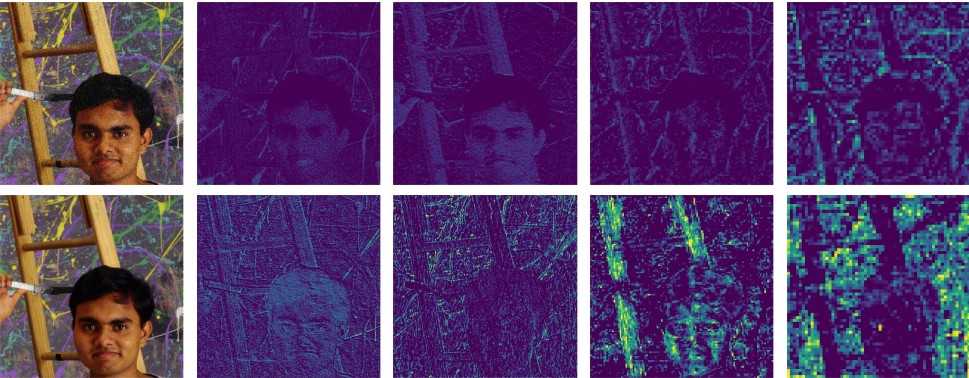

Figure 6: Qualitative comparisons on the intermediate multi-scale features in MSANet. The top left is the noise image, and the bottom left is the corresponding clean image. From left to right, the feature resolution varies from high to low. The top row is the features before (*i.e.*, without) subnetworks, and the bottom row is the features after (*i.e.*, with) subnetworks.

of multi-scale architecture design, we will explore other solutions to simultaneously exploit the within-scale characteristics and the cross-scale complementarity, and investigate their effectiveness in broader tasks such as deblur, segmentation.

**Broader Impact Statement.** MSANet is a specifically designed architecture for supervised single image denoising, which requires intensive labor to collect a lot of noisy-clean image pairs, and thus has the potential to make more opportunities for employment. However, MSANet is a general neural network and might be trained with uncertain data and used for uncertain purposes, such as watermark removal, which might prejudice the rights of others. Besides, MSANet involves a novel idea of multi-scale architecture design and might be used to design new networks for uncertain purposes. Moreover, the training and running of the model consume a lot of electricity causing carbon emission.

## Acknowledgements

This work was supported in part by NSFC under Grant U21B2040, U19A2078, and 61836006; and in part by Sichuan Science and Technology Planning Project under Grant 2022YFQ0014.

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
