# Supplementary Material
# Multi-Scale Adaptive Network
# for Single Image Denoising

**Yuanbiao Gou**[1]**, Peng Hu**[1]**, Jiancheng Lv**[1]**, Joey Tianyi Zhou**[2]**, Xi Peng**[1]*

[1]College of Computer Science, Sichuan University, China.
[2]Institute of High Performance Computing, A*STAR, Singapore.
{gouyuanbiao, penghu.ml, joey.tianyi.zhou, pengx.gm}@gmail.com
lvjiancheng@scu.edu.cn

## 1  Experiments

In this material, we present full implementation details and qualitative results of MSANet, including model configurations, model complexity, and additional visual comparisons on evaluation datasets.

### 1.1  Model Configurations

As mentioned in the main body of the paper, the proposed MSANet is built based on the residual block (RB), adaptive feature block (AFeB), adaptive multi-scale block (AMB), and adaptive fusion block (AFuB). To be specific, RB contains a convolutional layer with either (kernel size $K = 3$, stride $S = 2$, padding $P = 1$) or ($K = 1$, $S = 1$, $P = 0$) in terms of whether to downsample, following two convolutional layers ($K = 3$, $S = 1$, $P = 1$) with a LeakyReLU layer (negative slope $NS = 0.2$) between them and a skip-connection over them. As the architectures of AFeB, AMB and AFuB have been elaborated in the paper, we supplement some details here. (1) The convolutional and LeakyReLU layers in AFeB, AMB, AFuB adopt ($K = 3$, $S = 1$, $P = 1$, $NS = 0.2$) unless otherwise stated. (2) AMB contains four dilated convolutional layers ($K = 3$, $S = 1$) with the dilated rates of 1, 2, 3, 4, respectively. The convolutional layer used in adaptive feature scaling adopts ($K = 7$, $S = 1$, $P = 3$). (3) AFuB replaces the transpose convolutional layer ($K = 3$, $S = 2$, $P = 1$) with a standard convolutional layer ($K = 1$, $S = 1$, $P = 0$) when the upsampling operation is not carried.

### 1.2  Model Complexity

Here, we evaluate the model complexity of MSANet including parameter numbers, running time, and floating-point operations (FLOPs), and compare it with other methods. As shown in Table 1, although MSANet is not competitive in parameter numbers due to multiple subnetworks, its running time and FLOPs are attractive due to multi-resolution features.

### 1.3  Qualitative Comparisons

In addition to the results presented in the main body of the paper, we show more qualitative comparisons on evaluation datasets, i.e., SIDD (Fig. 1), Nam (Fig. 2), DnD (Fig. 3), CMcMaster (Fig. 4), CKodak24 (Fig. 5), CBSD68 (Fig. 6), GMcMaster (Fig. 7), GKodak24 (Fig. 8), and GBSD68 (Fig. 9). From these figures, one could observe that our MSANet achieves promising and visually pleasant denoising results. Some areas are highlighted by color rectangles and zooming-in is recommended for better visualization.

---

*Corresponding author

36th Conference on Neural Information Processing Systems (NeurIPS 2022).

Table 1: Comparisons of parameter numbers, running time, and FLOPs on $480 \times 320$ color image.

| Methods | Params (M) | Time (ms) | FLOPs (G) |
|---|---|---|---|
| DnCNN | 0.558 | 21.3 | 86.1 |
| RIDNet | 1.499 | 84.4 | 230.0 |
| SADNET | 4.321 | 26.7 | 50.1 |
| DeamNet | 1.873 | 99.2 | 342.3 |
| RNAN | 8.960 | 1072.2 | 1163.5 |
| MSANet | 7.997 | 94.7 | 166.2 |

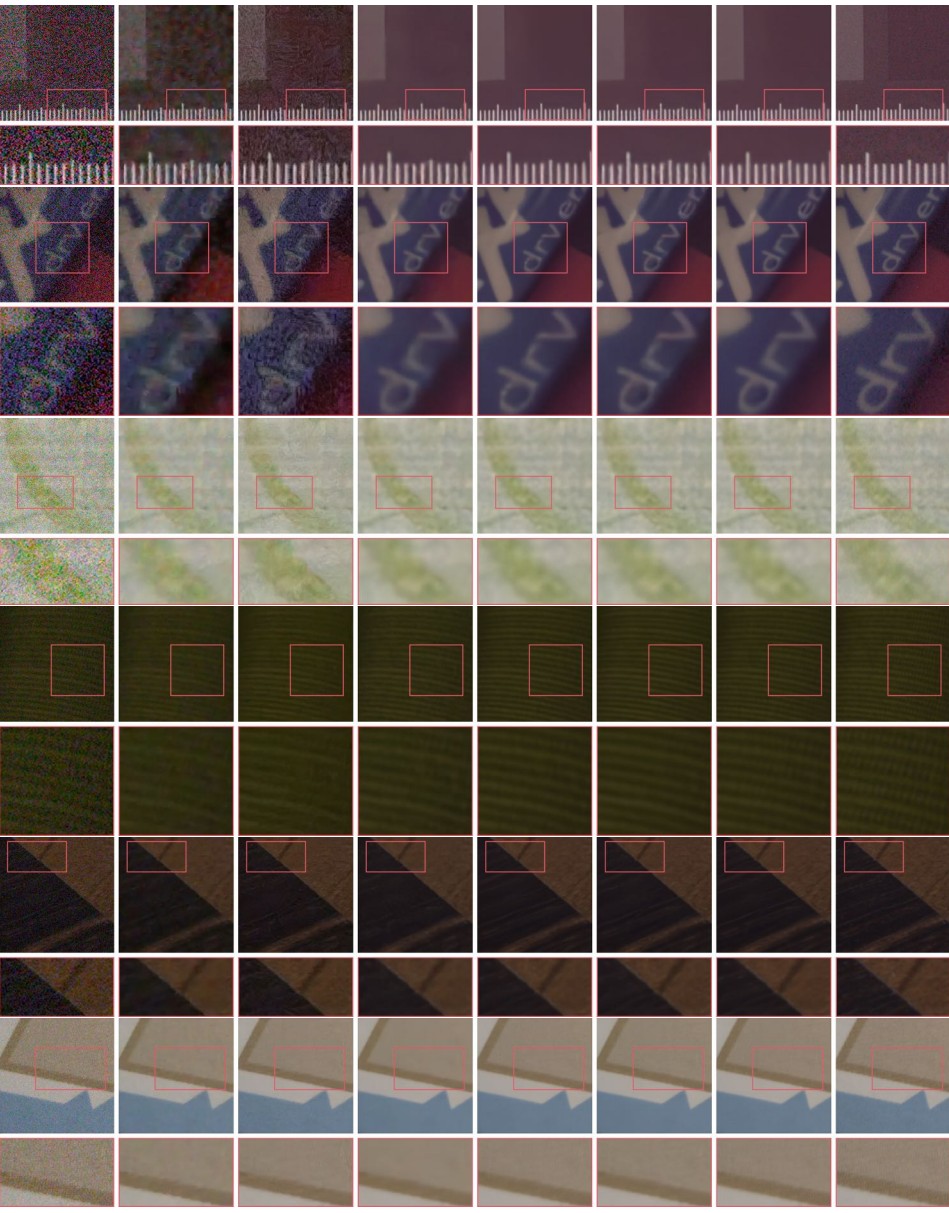

Figure 1: Qualitative comparisons on the SIDD dataset with real noise. From left to right, we show the real noise image, the results of CBDNet, PD, RIDNet, SADNet, DeamNet, MSANet, and the ground truth.

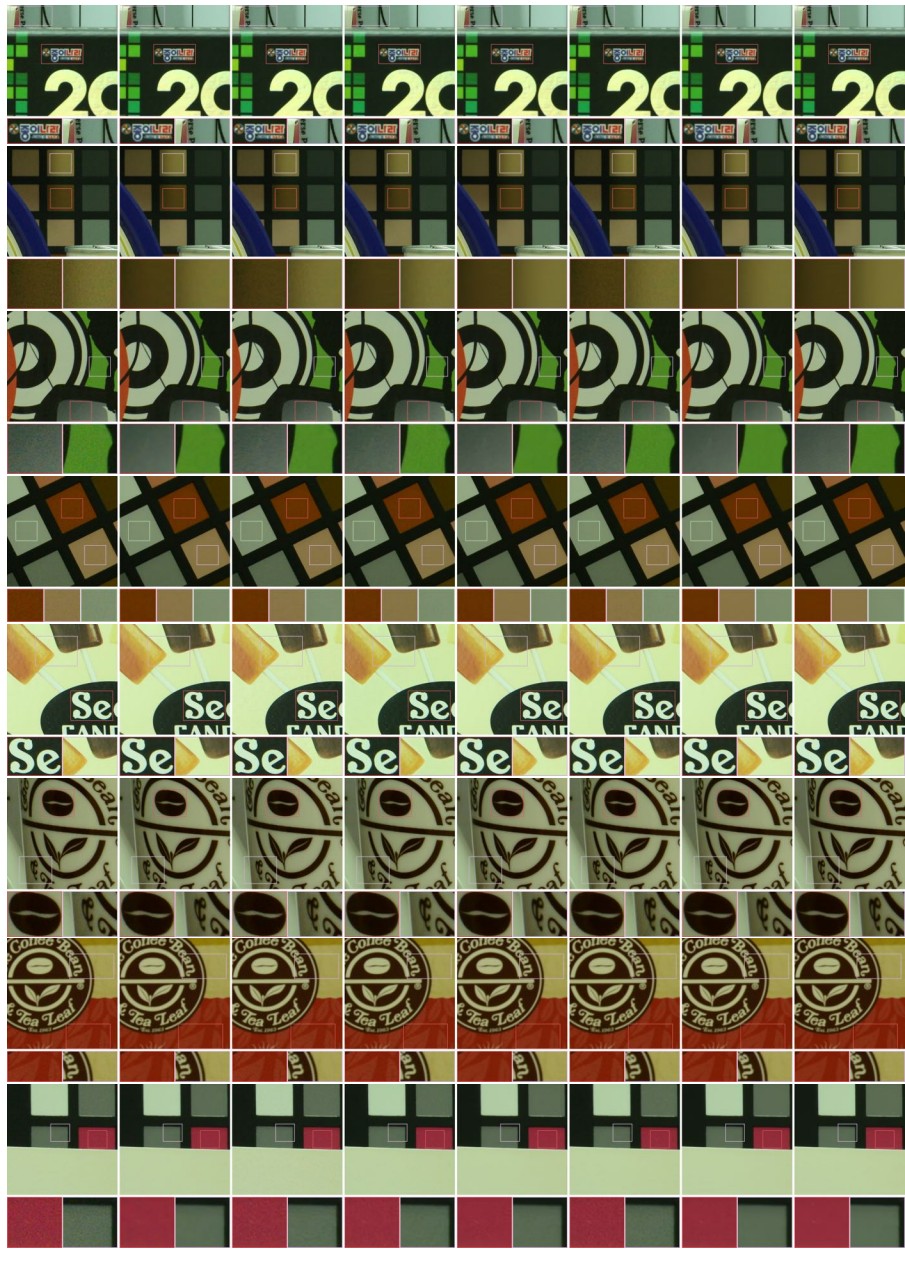

Figure 2: Qualitative comparisons on the Nam dataset with real noise. From left to right, we show the real noise image, the results of CBDNet, PD, RIDNet, SADNet, DeamNet, MSANet, and the ground truth.

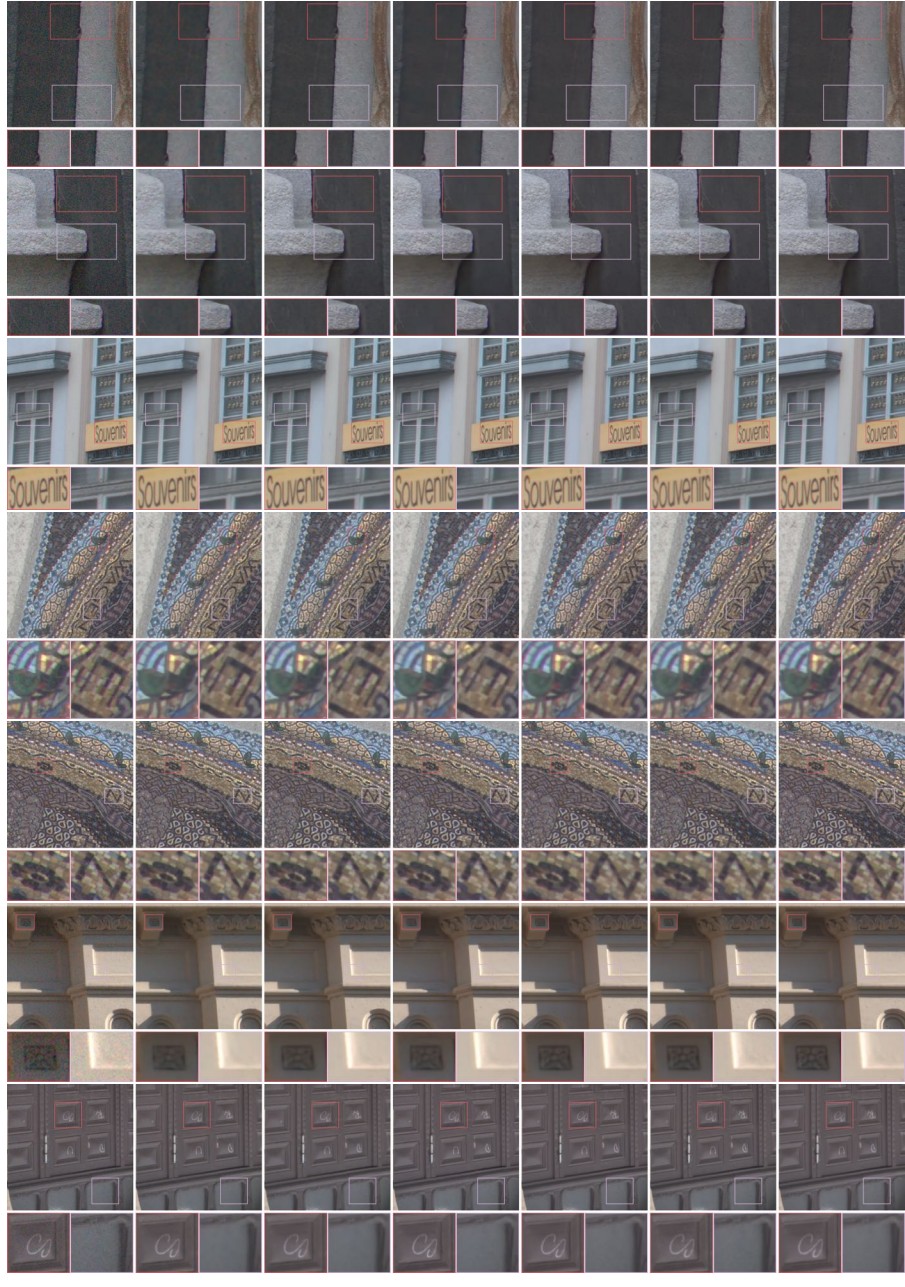

Figure 3: Qualitative comparisons on the DnD dataset with real noise. From left to right, we show the real noise image, the results of CBDNet, RIDNet, PD, SADNet, DeamNet, and MSANet.

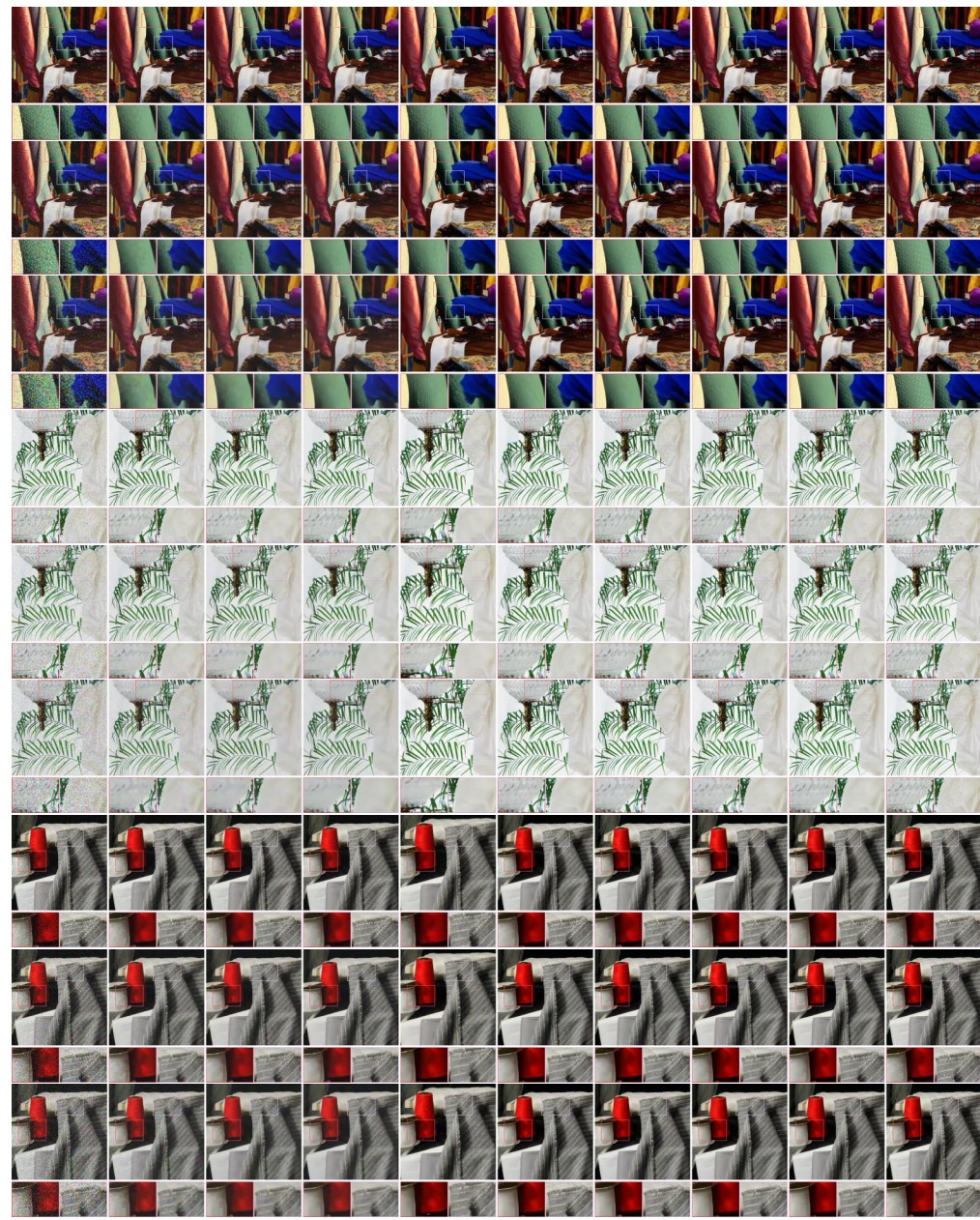

Figure 4: Qualitative comparisons on the color McMaster dataset with synthetic noise. From top to bottom of the same scene, the noise levels respectively are 30, 50, 70. From left to right, we show the synthetic noise image, the results of BM3D, DnCNN, FFDNet, CLEARER, SADNet, RNAN, DeamNet, MSANet, and the ground truth.

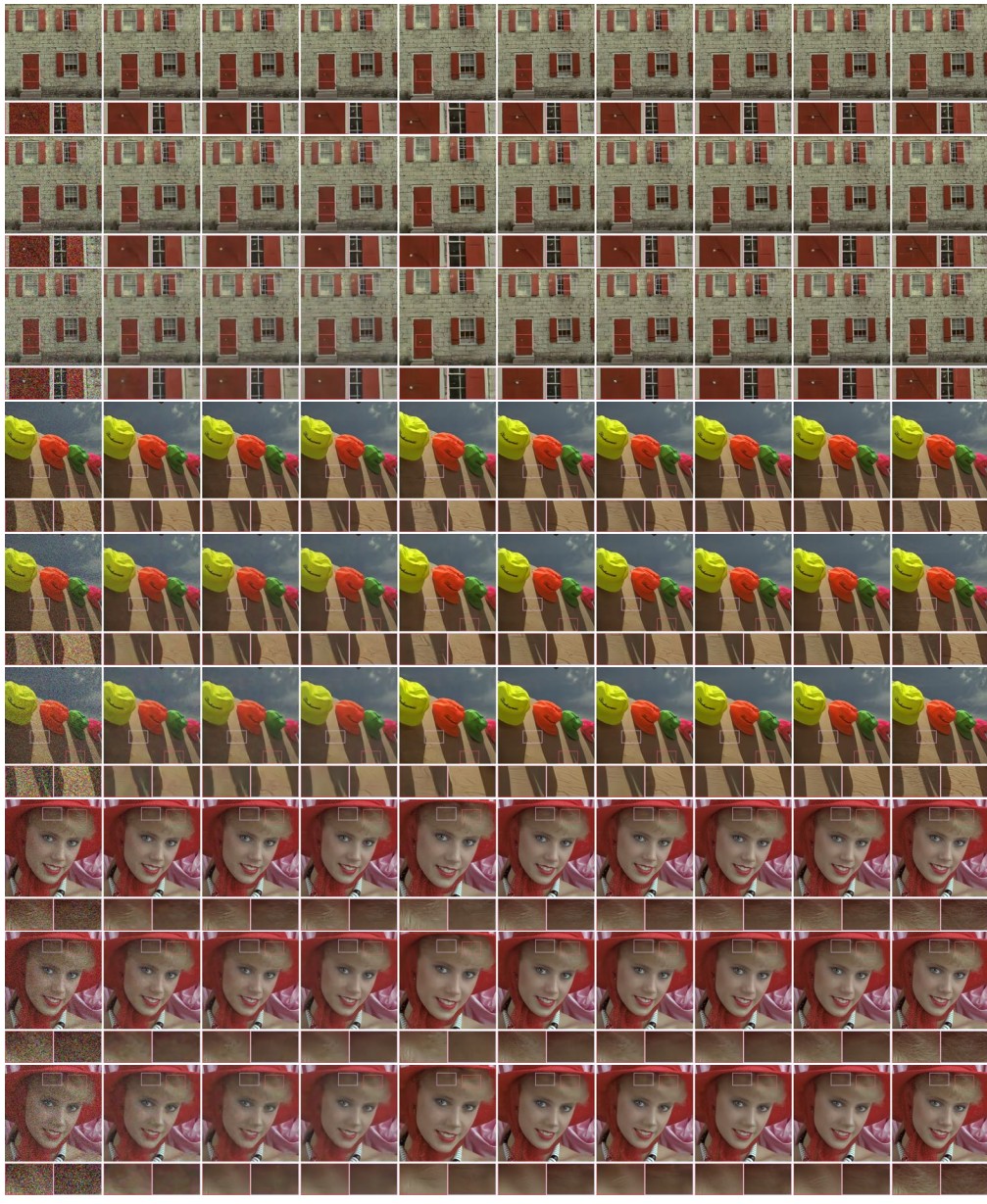

Figure 5: Qualitative comparisons on the color Kodak24 dataset with synthetic noise. From top to bottom of the same scene, the noise levels respectively are 30, 50, 70. From left to right, we show the synthetic noise image, the results of BM3D, DnCNN, FFDNet, CLEARER, SADNet, RNAN, DeamNet, MSANet, and the ground truth.

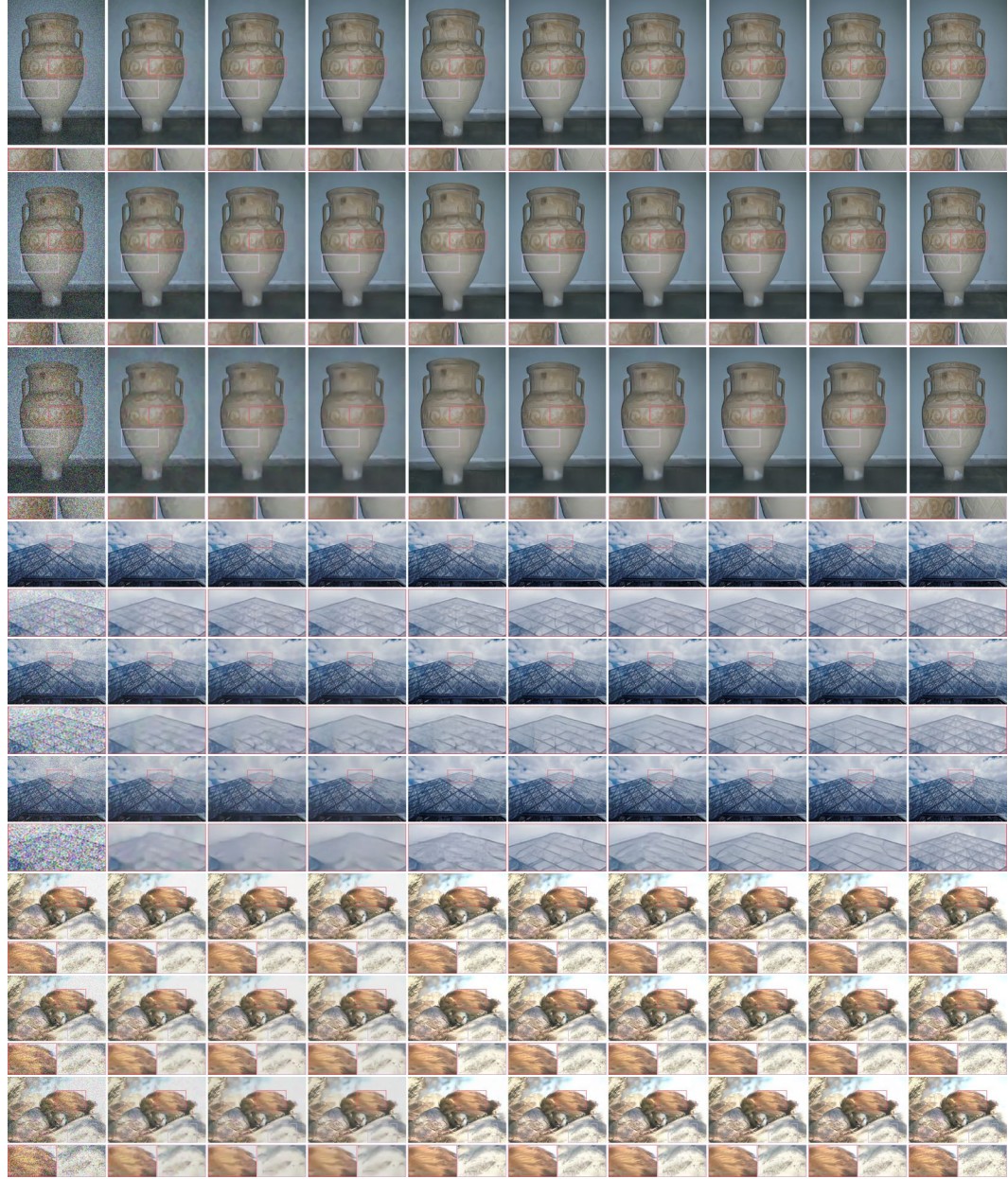

Figure 6: Qualitative comparisons on the color BSD68 dataset with synthetic noise. From top to bottom of the same scene, the noise levels respectively are 30, 50, 70. From left to right, we show the synthetic noise image, the results of BM3D, DnCNN, FFDNet, CLEARER, SADNet, RNAN, DeamNet, MSANet, and the ground truth.

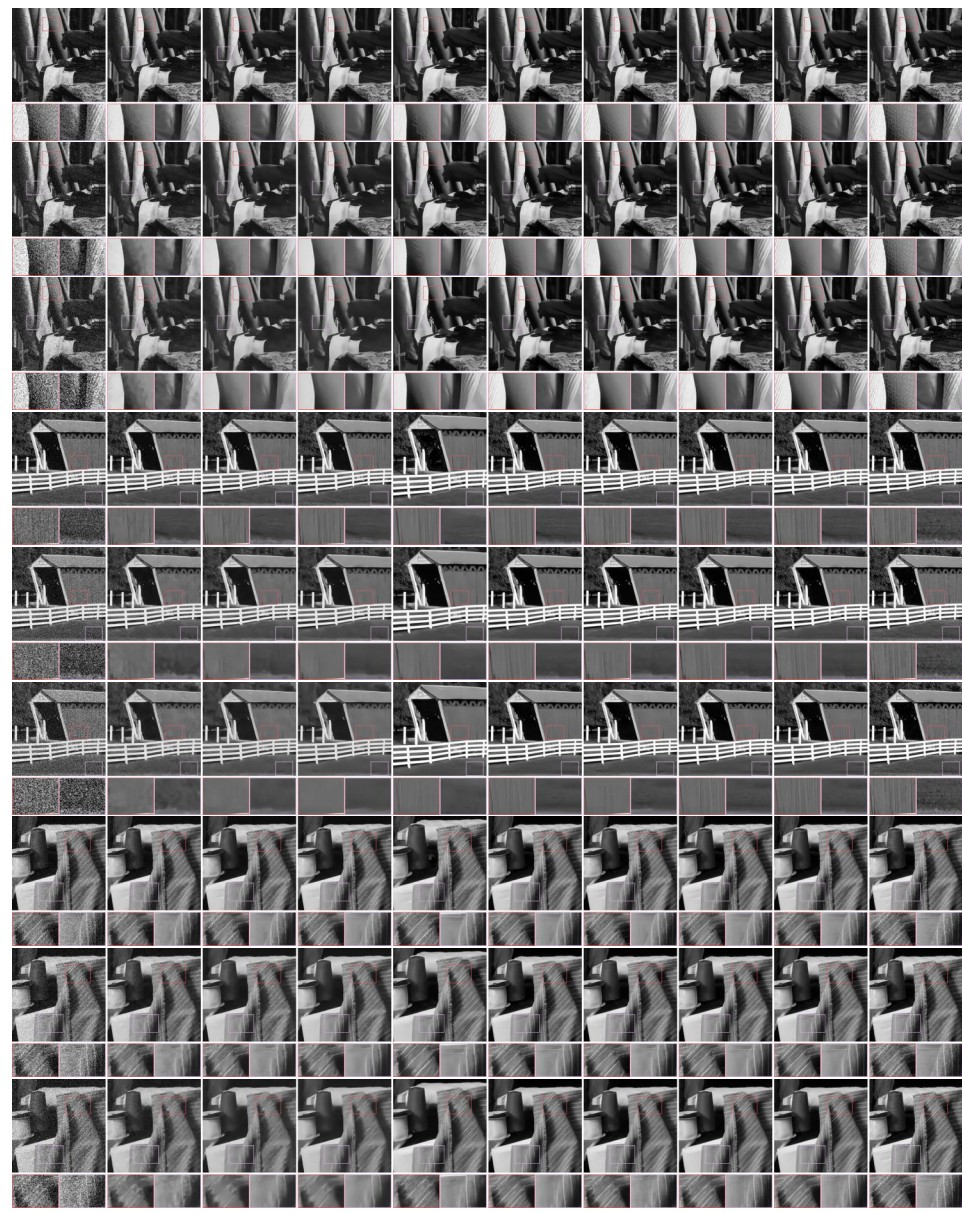

Figure 7: Qualitative comparisons on the grayscale McMaster dataset with synthetic noise. From top to bottom of the same scene, the noise levels respectively are 30, 50, 70. From left to right, we show the synthetic noise image, the results of BM3D, DnCNN, FFDNet, CLEARER, SADNet, RNAN, DeamNet, MSANet, and the ground truth.

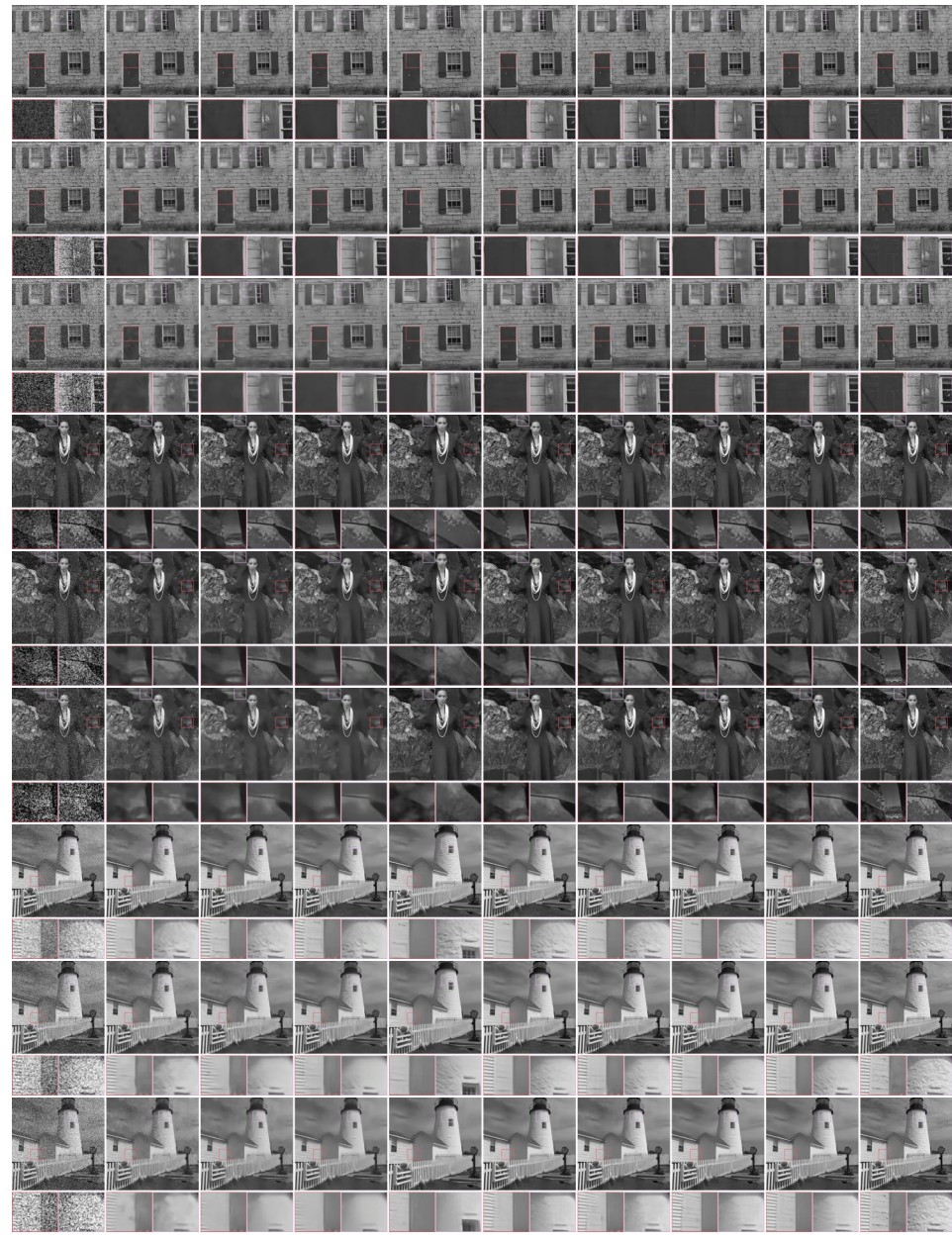

Figure 8: Qualitative comparisons on the grayscale Kodak24 dataset with synthetic noise. From top to bottom of the same scene, the noise levels respectively are 30, 50, 70. From left to right, we show the synthetic noise image, the results of BM3D, DnCNN, FFDNet, CLEARER, SADNet, RNAN, DeamNet, MSANet, and the ground truth.

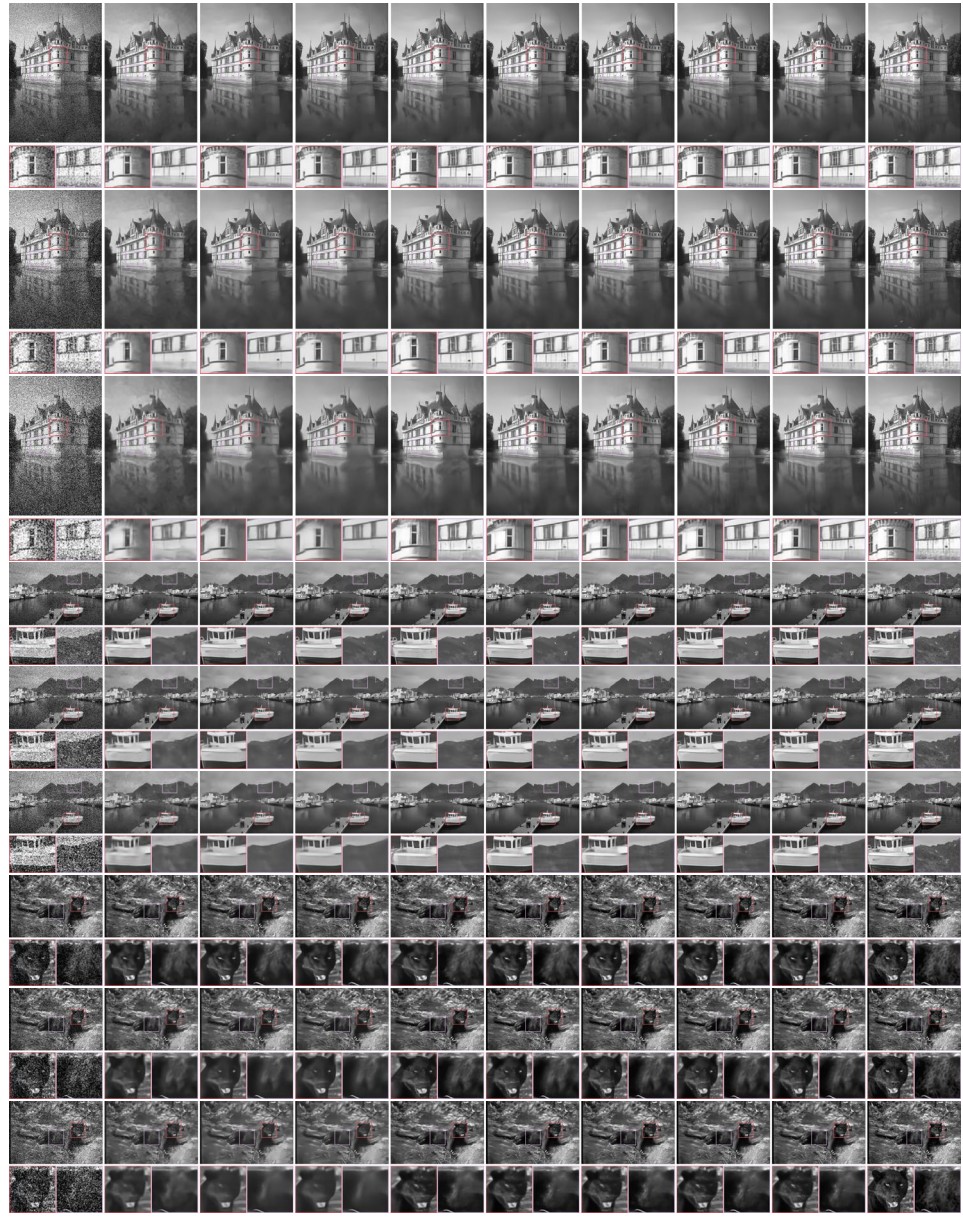

Figure 9: Qualitative comparisons on the grayscale BSD68 dataset with synthetic noise. From top to bottom of the same scene, the noise levels respectively are 30, 50, 70. From left to right, we show the synthetic noise image, the results of BM3D, DnCNN, FFDNet, CLEARER, SADNet, RNAN, DeamNet, MSANet, and the ground truth.