# OpenReview forum: "Multi-Scale Adaptive Network for Single Image Denoising"
_NeurIPS.cc/2022/Conference — NeurIPS 2022 Accept_

### Official Review · Reviewer_H79u · 2022-07-04

**Rating:** 7
**Confidence:** 4
**Soundness:** 3 good
**Presentation:** 4 excellent
**Contribution:** 3 good

**Summary:**

Different from existing works treat different scale features equally, in this paper, the authors reveal the different scale features show varying characteristics and should be processed by scale-speciﬁc rather than homologous structures. To simultaneously embrace the within-scale characteristics and cross-scale complementarity of multi-scale features, the authors propose a novel Multi-Scale Adaptive Network (MSANet) for single image denoising. Speciﬁcally, three neural blocks are designed, i.e., adaptive feature block (AFeB) for adaptively sampling and ﬁltering features; adaptive multi-scale block (AMB) for expanding the receptive ﬁeld and adaptively aggregate multi-scale features; adaptive fusion block (AFuB) for adaptively fusing the multi-scale features with varying characteristics. Ablation studies indicate the blocks are eﬀective, and comparison experiments show MSANet achieves better performance than alternatives.


**Questions:**

1. In the paper, “adaptive” is frequently used to describe the proposed network as well as the three neural blocks, but itself is not explicitly discussed. Some discussions are recommended for a better understanding to this work.
2. AFeB is designed to adaptively sampling and weighting the input features, which is highly expected for ﬁne-grained features. However, why adaptively sampling and weighting the ﬁne-grained features is good for the denoising task to preserve the image details and filter unpleasant noise?
3. As Table 6 suggests, the performance gains of using either AFeB or AMS alone are slight. However, why using them together could bring significant performance improvements? Please give some interpretations.


**Limitations:**

Yes, the authors have discussed the related problems.

**Strengths And Weaknesses:**

Strengths:
1. The authors reveal the different scale features show varying characteristics and should be processed by scale-speciﬁc structures for this first time. Based on the observation, MSANet is proposed to utilize the within-scale characteristics and cross-scale complementarity of multi-scale features. Extensive experiments are conducted on both three real and six synthetic noisy image datasets to show its superiority.
2. Three neural blocks are designed by considering the within-scale characteristics and cross-scale complementarity of multi-scale features. Ablation studies indicate these blocks are eﬀective.
3. In general, the paper is well written. The related works are clearly illustrated, the shortcomings of existing multi-scale networks are analyzed, and accordingly MSANet is proposed by solving the shortcomings. The framework is clear and easy to understand, and the experimental results are explained in detail.

Weaknesses:
1. The idea of taking advantage of within-scale characteristics and cross-scale complementarity is not limited to denoising task, but is a general idea of multi-scale architecture design. Therefore, exploring and verifying this idea in more tasks and areas would make this work be more significant.
2. MSANet contains more parameters than most of the baselines due to multiple subnetworks for multi-scale features.

---

> ### Author Response · Authors · 2022-08-02
> **Response to Reviewer H79u**
>
> **Q1: The idea of taking advantage of within-scale characteristics and cross-scale complementarity is not limited to denoising task, but is a general idea of multi-scale architecture design. Therefore, exploring and verifying this idea in more tasks and areas would make this work be more significant.**
>
> **A1:** By analyzing the characteristics of multi-scale features w.r.t. noisy images, we reveal the **within-scale characteristics (WSC)** and naturally verify its effectiveness in denoising. Although our idea is general to other tasks, with limited time and resources, it is unnecessary to extend it to handle other corruptions in a paper because denoising is a severely ill-posed problem and one of the most important low-level vision tasks. We would highlight that there are two parallel research paradigms: i) highlighting the generality of the method w.r.t. different tasks; ii) diving into a given task and accordingly developing a general solution. Clearly, this study belongs to the latter and we believe it could provide sufficient insight to the community.
>
> **Q2: MSANet contains more parameters than most of the baselines due to multiple subnetworks for multi-scale features.**
>
> **A2:** Although MSANet is less attractive in parameters, its FLOPs are obviously lower than most baselines even with more parameters. Moreover, we would remind that one should pay more attention to the novelty and insight of this work to the community, i.e., the WSC of features are varying instead of fixed with the scales, which is our motivation for multi-scale architecture design and such a property is not reported so far as we known.
>
> **Q3: In the paper, “adaptive” is frequently used to describe the proposed network as well as the three neural blocks, but itself is not explicitly discussed.**
>
> **A3:** In the network level, the “adaptive” mainly refers to the capability of extracting scale-specific features and fuse them based on their characteristics, i.e., exploiting the WSC and the **cross-scale complementarity (CSC)**. To achieve the “adaptive”, the network should consider the characteristics of multi-scale features, and properly design and use the modules to adapt the scales’ characteristics. In the module level, the “adaptive” mainly refers to the capability of sampling and weighting the features based on themselves. With “adaptive”, the modules have the ability to automatically discriminate the indispensable input features from those unpleasant ones, and determine their contributions to output features. AFeB and AFuB employ deformable convolution while AMB uses dilated convolution and channel-spatial attention to achieve “adaptive”. Their differences of design are from the characteristics of multi-scale features they adapt to.
>
> **Q4: AFeB is designed to adaptively sampling and weighting the input features, which is highly expected for ﬁne-grained features. However, why adaptively sampling and weighting the ﬁne-grained features is good for the denoising task to preserve the image details and filter unpleasant noise?**
>
> **A4:** AFeB could learn the sampling locations to indicate where are important for recovery, while assigning different weights to show how important the locations are, based on the input features. As a result, AFeB could preserve the image details and filter unpleasant noise from the input features for better recovery performance.
>
> **Q5: As Table 6 suggests, the performance gains of using either AFeB or AMS alone are slight. However, why using them together could bring significant performance improvements?**
>
> **A5:** As one WSC of high-resolution features is the mixture of details and noises, AFeB is designed to exploit this characteristic for adaptively preserving the indispensable details and filtering unpleasant noises. As one WSC of low-resolution features is with rich contextual information but too low-resolution destroys the image contents, AMB is designed for enriching the contextual information while keeping the resolution unchanged. Therefore, using them together could fully exploit the WSC of multi-scale features, and thus achieve better performance.

---

> ### Author Response · Authors · 2022-08-06
> **To Reviewer H79u**
>
> Dear reviewer H79u,
>
> Thanks a lot for reviewing our paper and giving us the valuable suggestions.
>
> We have tried our best to answer all the questions according to the comments. We sincerely hope that our responses could address all your concerns. Is there anything that needs us to further clarify for the given concerns?
>
> Thanks again for your hard work.

---

> ### Comment · Reviewer_H79u · 2022-08-09
> **Post-rebuttal**
>
> I would like to thank the authors for their response to my review, which have addressed my concerns. In consideration of the response to me and other reviewers, I think the revealed issue is novel, and the proposed solution is effective, which has the potential of bringing insights and inspirations to multi-scale architecture design. Therefore, I would keep my rating and recommend the paper be accepted.

---

> > ### Author Response · Authors · 2022-08-09
> > **Response to Post-rebuttal**
> >
> > Thanks for your positive comments and suggestions. We would improve our manuscript for a clearer presentation in the next version.

---

### Official Review · Reviewer_9KYY · 2022-07-07

**Rating:** 7
**Confidence:** 4
**Soundness:** 3 good
**Presentation:** 3 good
**Contribution:** 3 good

**Summary:**

To exploit the potential of the within-scale characteristics and cross-scale complementarity of multi-scale features, three elaborate neural blocks and a novel Multi-Scale Adaptive Network (MSANet) are proposed for single image denoising. Specifically, AFeB and AMB are designed by taking the within-scale characteristics of multi-scale features into consideration, AFuB is designed to exploit the cross-scale complementarity of multi-scale features. Three neural blocks are combined to be the scale-wise subnetwork in MSANet by adapting to the feature characteristics of the corresponding scale. Ablation studies demonstrate the effectiveness of the proposed neural blocks and network. Sufficient comparison experiments on three real and six synthetic noisy image datasets compared with 12 baselines show the advantages of the proposed method.

**Questions:**

1. Although different-scale features are processed by different subnetworks, and the subnetworks in the finest and coarsest scale are clearly scale-specific. However, it’s difficult to understand the scale-specific designs in the two middle subnetworks. More clear interpretations are needed for a better understanding.
2. As shown in Table 6, using AFeB and AMS together could signiﬁcantly improve the performance. However, using either AFeB or AMB alone slightly gains the performance over ResB. Why is that? Some clear explanations are needed for a better understanding.
3. As shown in Table 7, although MSANet contains more parameters, the running time and the FLOPs are not much. Why does this happen? Please give some explanations.
4. Some typos, e.g., “disorderly” -> “disordered” in the paragraph of Adaptive Fusion Block (AFuB).


**Limitations:**

Yes

**Strengths And Weaknesses:**

A. Strengths
1. The idea of simultaneously exploiting the within-scale characteristics and cross-scale complementarity of multi-scale features is interesting and reasonable. The accordingly proposed MSANet incorporates them into multi-scale architecture design, and shows impressive performance on single image denoising task.
2. The design of AFuB is novel and ingenious. Taking the disordered ﬁne-grained image details in high-resolution features and the coarse-grained image context in low-resolution features as input to adaptively sample, weight and transfer the ﬁne-grained image details into the coarse-grained image context.
3. Sufficient experiments are conducted to demonstrate the effectiveness of the proposed designs and the advantages over the previous works. Moreover, the supplementary material also shows more details and results together with the main paper.

B. Weaknesses
1. Since the cross-scale complementarity and the within-scale characteristics are not only restricted to single image denoising task. Therefore, the designs will be more persuasive if they could also achieve improvements in other image restoration tasks.
2. Compared with the three real and the three synthetic color noisy image datasets, the performance improvements over the baselines on the three synthetic grayscale noisy image datasets are less significant.

---

> ### Author Response · Authors · 2022-08-02
> **Response to Reviewer 9KYY**
>
> **Q1: Since the cross-scale complementarity and the within-scale characteristics are not only restricted to single image denoising task. Therefore, the designs will be more persuasive if they could also achieve improvements in other image restoration tasks.**
>
> **A1:** We would clarify that **denoising is one of the most important restoration tasks and a severely ill-posed problem.** By analyzing the characteristics of multi-scale features w.r.t. noisy images, we reveal the **within-scale characteristics (WSC)** and naturally verify its effectiveness in denoising. Although the idea is general, we do not think it is necessary to verify its effectiveness to other applications in a conference paper. In fact, we would highlight that there are two equally important research paradigms: i) highlighting the generality of the method w.r.t. different tasks; ii) diving into a given task and accordingly developing a general solution. Clearly, this study belongs to the latter and we believe it could provide sufficient insight to the community.
>
> **Q2: Compared with the three real and the three synthetic noisy image datasets, the performance improvements over the baselines on the three synthetic grayscale noisy image datasets are less significant.**
>
> **A2:** First, the focus of the community has gradually shifted to real-world noise, on which our method achieves considerable improvements (See Table 1-3 in manuscript). Second, even on the synthetic noise, it is inaccurate to say the improvement is limited. By referring to the performance gaps between these SOTAs (SADNet, RNAN, DeamNet), one could see that our improvements are reasonable, e.g., the first row in Table 4, the PSNR/SSIM gap between the best two baselines is 0.01dB/0.0005, while the improvement of our method over the best baseline is 0.06dB/0.0014. Similar observations could also be obtained in other rows.
>
> **Q3: Although different-scale features are processed by different subnetworks, and the subnetworks in the finest and coarsest scale are clearly scale-specific. However, it’s difficult to understand the scale-specific designs in the two middle subnetworks.**
>
> **A3:** As the characteristics of multi-scale features gradually changes from high- to low-resolution, for simplicity, we take the two bottom and two top resolutions in the Fig.2 of the paper as the high- and low-resolution, respectively. For high-resolution features, AFeB and AMB are alternately used to exploit their WSC. For low-resolution features, AMB is used to exploit their WSC. Meanwhile, except for the lowest resolution, the first and the last blocks in each subnetwork are AFeB for adaptively selecting the input and output features of each subnetwork. Following the above architecture principles, the two middle subnetworks are designed as shown in the Fig.2.
>
> **Q4: As shown in Table 6, using AFeB and AMS together could signiﬁcantly improve the performance. However, using either AFeB or AMB alone slightly gains the performance over ResB. Why is that? Some clear explanations are needed for a better understanding.**
>
> **A4:** AFeB and AMB together exploit the WSC. As one WSC of high-resolution features is the mixture of details and noises, AFeB is designed to exploit this characteristic for adaptively preserving the indispensable details and filtering unpleasant noises. As one WSC of low-resolution features is with rich contextual information while a too low-resolution destroys the image contents, AMB is designed for enriching the contextual information while keeping the resolution unchanged. Therefore, suboptimal results will be obtained if using either AFeB or AMB alone, i.e., the WSC of multi-scale features is partially neglected.
>
> **Q5: As shown in Table 7, although MSANet contains more parameters, the running time and the FLOPs are not much. Why does this happen? Please give some explanations.**
>
> **A5:** Modern multi-scale architectures usually consist of multiple stages. At the end of each stage, the feature resolution will be halved while the feature channels will be doubled. As a result, the parameters will increase due to the doubled channels, and the FLOPs and running time will decrease due to the halved resolution (Height & Width).
>
> **Q6: Some typos, e.g., “disorderly” -> “disordered” in the paragraph of Adaptive Fusion Block (AFuB).**
>
> **A6:** We will revise the typos and carefully reinspect the writing in the next version.

---

> ### Author Response · Authors · 2022-08-06
> **To Reviewer 9KYY**
>
> Dear reviewer 9KYY,
>
> Thanks a lot for reviewing our paper and giving us the valuable suggestions.
>
> We have tried our best to answer all the questions according to the comments. We sincerely hope that our responses could address all your concerns. Is there anything that needs us to further clarify for the given concerns?
>
> Thanks again for your hard work.

---

> > ### Comment · Reviewer_9KYY · 2022-08-08
> > **Post-rebuttal**
> >
> > Thanks to the authors for the detailed responses. My concerns have been well solved.
> > I keep my initial rating to the submission.

---

> > > ### Author Response · Authors · 2022-08-09
> > > **Response to Post-rebuttal**
> > >
> > > Thanks for your positive comments and suggestions. We would accordingly revise the problems and include some discussions about the concerns in the next version for a clearer presentation.

---

### Official Review · Reviewer_D7k7 · 2022-07-10

**Rating:** 7
**Confidence:** 4
**Soundness:** 3 good
**Presentation:** 3 good
**Contribution:** 3 good

**Summary:**

Overall, this paper aims to tackle the task of image denoising based on the drawback (not exploring the internal information of each scale in the network) of multi-scale networks. Specifically, the authors propose three modules to extract feature maps, multi-scale construction and feature aggregation to run a noisy image. Extensive experimental results are shown in the paper.

**Questions:**

[-] My problem is mainly how can the authors prove that the existing approach does not make use of the information within the scale?

[-] The authors' proposed approach does not seem to be innovative.

**Limitations:**

Limitations have been illustrated.

**Strengths And Weaknesses:**

Overall, the architecture of the paper is clear and the writing is good, but the theoretical and methodological parts are not enough to meet me.

Strengths

[+]  The authors seem to propose an effective image denoising solution, described in the motivation, related work, methodology and experiments.

Weaknesses

[-] First of all, the title does not show the author's intention, which is a bad case (No expressed motivation).

[-] The abstract description starts by depicting a big problem, namely the general weakness of multi-scale networks, but focuses on a simple application.

[-] The introduction and related works ignore Transformer-based models.

[-] Figure 1 does not convey the author's intention, which is confusing. Why not use the feature map presentation to illustrate that there is a problem with information extraction within the scale?

[-] Do not insert ‘so on’ to follow ‘such as’.

[-] The biggest problem is that the motivation is not valid and I don't see cases to illustrate the problems with multi-scale networks nowadays.

[-] In addition, the solution is essentially adding convolutional layers (adaptive) between scales, does this solve the problem?

[-] What worries me is that the methods of CVPR'22 have been released, so why are there no relevant experiments to compare these methods?

[-] relu---> ReLU, pytorch--->PyTorch.

---

> ### Author Response · Authors · 2022-08-02
> **Response to Reviewer D7k7 (Part 1)**
>
> **Q1: First of all, the title does not show the author's intention, which is a bad case (No expressed motivation).**
>
> **A1:** The authors think the title is good, since it clearly conveys “the task” and “the solution” of this paper. Moreover, the “Adaptive” encapsulates the motivations and the characteristic of our solution.
>
> **Q2: The abstract description starts by depicting a big problem, namely the general weakness of multi-scale networks, but focuses on a simple application.**
>
> **A2:** First, we sincerely thank for the recognition on one major contribution of this work, i.e., pointed out a general weakness of multi-scale networks. Second, we would remind that **denoising is one of the most important restoration tasks and not a simple application.** By analyzing the characteristics of multi-scale features w.r.t. noisy images, we reveal the **within-scale characteristics (WSC)** and naturally verify its effectiveness in denoising. Although the idea is general, we do not think it is necessary to verify its effectiveness to other applications in a conference paper. In fact, we would remind that there are two equally important research paradigms: i) highlighting the generality of the method w.r.t. different tasks; ii) diving into a given task and accordingly developing a general solution. Clearly, this study belongs to the latter and we believe it could provide sufficient insight to the community.
>
> **Q3: The introduction and related works ignore Transformer-based models.**
>
> **Q5: Do not insert ‘so on’ to follow ‘such as’.**
>
> **Q9: relu---> ReLU, pytorch--->PyTorch.**
>
> **A3, A5 and A9:** We will revise the problems and carefully reinspect the writing in the next version.
>
> **Q4: Figure 1 does not convey the author's intention, which is confusing. Why not use the feature map presentation to illustrate that there is a problem with information extraction within the scale?**
>
> **A4:** For a more concise and clearer illustration on our motivation, Fig.1 takes a more abstract fashion which is more like a NeurIPS paper. Compared to the raw multi-scale feature maps, such a summarized and abstracted illustration is more conducive to readers to understand. In the new Supplementary Material, we show the qualitative results of intermediate features before (i.e., without) and after (i.e., with) all MSANet subnetworks to demonstrate the significance of exploiting WSC.
>
> **Q6: The biggest problem is that the motivation is not valid and I don't see cases to illustrate the problems with multi-scale networks nowadays.**
>
> **A6:** We believe that the quantitative and qualitative results have sufficiently verified the effectiveness of the motivation and our design criterion. To further improve the work, the new supplementary material includes the qualitative results of intermediate features before (i.e., without) and after (i.e., with) our subnetworks, and we believe it could be another evidence to qualitatively demonstrate the significance of exploiting WSC.
>
> Moreover, we sincerely hope the reviewer could take the following questions into consideration and answer them if possible, i) are there any publications revealed the same phenomenon / weakness of multi-scale network? i.e., different scale features show varying characteristics and should be processed by scale-speciﬁc structures rather than homologous architectures. ii) do our ablation studies and related experimental results not verify the effectiveness of our solution? (See A3 to Reviewer rLGu for a more detailed explanation); iii) does this work not provide novel insight to the community?
>
> **Q7: In addition, the solution is essentially adding convolutional layers (adaptive) between scales, does this solve the problem?**
>
> **A7:** This does solve the problem. Like deep neural network, although what they essentially do is linear and nonlinear transformations, there are a lot of really brilliant designs such as Transformers, CNNs, RNNs, LSTMs. Namely, the most important is not what are used, but how are they used and what problems are solved by them, in which problems are the most important in scientific research. Obviously, this paper first reveals the problem in modern multi-scale architecture designs, and then proposes the novel designs to solve this problem, finally, evaluates its effectiveness on denoising which is one of the most important restoration tasks and a severely ill-posed problem.

---

> > ### Author Response · Authors · 2022-08-02
> > **Response to Reviewer D7k7 (Part 2)**
> >
> > **Q8: What worries me is that the methods of CVPR'22 have been released, so why are there no relevant experiments to compare these methods?**
> >
> > **A8:** We would remind the reviewer that **the papers of CVPR'22 were released in June, while the Paper Submission Deadline of NeurIPS'22 is in May**. Besides, we mainly select the baselines that are comparable to our methods in parameters and FLOPs. Taking an image of 256$\times$256 as example, the parameters/FLOPs of Restormer[1], Uformer[2], DGUNet[3] respectively are 26M/282G, 51M/179G, 17M/1729G, which are far more than our MSANet which is 8M/71G.
> >
> > [1] Zamir, Syed Waqas, et al. Restormer: Efficient transformer for high-resolution image restoration. In Proceedings of the IEEE/CVF Conference on Computer Vision and Pattern Recognition. 2022.
> >
> > [2] Wang, Zhendong, Xiaodong Cun, Jianmin Bao, Wengang Zhou, Jianzhuang Liu, and Houqiang Li. Uformer: A general u-shaped transformer for image restoration. In Proceedings of the IEEE/CVF Conference on Computer Vision and Pattern Recognition. 2022.
> >
> > [3] Mou, Chong, Qian Wang, and Jian Zhang. Deep Generalized Unfolding Networks for Image Restoration. In Proceedings of the IEEE/CVF Conference on Computer Vision and Pattern Recognition. 2022.
> >
> > **Q10: My problem is mainly how can the authors prove that the existing approach does not make use of the information within the scale?**
> >
> > **A10:** We did not claim that existing methods are “not exploring the internal information of each scale in the network” and hence accordingly we do not need to prove that ``the existing approach does not make use of the information within the scale’’. In this paper, the revealed drawback of multi-scale networks is that they ignore the within-scale characteristics (WSC) in architecture design, i.e., different scale features show varying characteristics but existing multi-scale networks use homologous architectures to deal with them. To the best of our knowledge, such a drawback is revealed for the first time, and if the reviewer sees the same works, kindly let us to know please.
> >
> > **Q11: The authors' proposed approach does not seem to be innovative.**
> >
> > **A11:** Novelty is the quality of being new, or following from that, of being striking, original or unusual. In this paper, we reveal a new problem in modern multi-scale architecture designs, and accordingly proposes new design principles, adaptive modules and network. Besides, the effectiveness has also been verified in experiments. I don't understand why the reviewer thinks they are not innovative. Reviewer should provide materials to support his statements, such as the previous works that are same to our work in observations/motivations, design principles, adaptive modules as well as network (see our questions in A6).

---

> ### Author Response · Authors · 2022-08-06
> **To Reviewer D7k7**
>
> Dear reviewer D7k7,
>
> Thanks a lot for reviewing our paper and giving us the valuable suggestions.
>
> We have tried our best to answer all the questions according to the comments. We sincerely hope that our responses could address all your concerns. Is there anything that needs us to further clarify for the given concerns?
>
> Thanks again for your hard work.

---

### Official Review · Reviewer_9gQL · 2022-07-10

**Rating:** 7
**Confidence:** 5
**Soundness:** 4 excellent
**Presentation:** 4 excellent
**Contribution:** 3 good

**Summary:**

The paper investigates existing multi-scale methods, and discovers the within-scale characteristics of multi-scale features are ignored. Therefore, the paper reveals this missing piece for multi-scale architecture design, and accordingly proposes a novel Multi-Scale Adaptive Network (MSANet) by simultaneously exploiting the within-scale characteristics and the cross-scale complementarity. MSANet uses AFeB, AMB to build different subnetworks corresponding to diﬀerent scales for exploiting the within-scale characteristics, and AFuB to fuse the multi-scale features with varying characteristics from coarse to ﬁne for exploiting the cross-scale complementarity. Extensive experiments on both three real and six synthetic noisy image datasets show the effectiveness of the proposed designs and the advantages over the previous methods. Moreover, supplementary material shows more results together with the main paper.

**Questions:**

(1) Both AFeB and AFuB are based on deformable convolution to adaptively sample and weight the features, but one is used to adaptively select details and ﬁlter noises, the other is used to fuse the multi-scale features with varying characteristics. What caused this crucial difference? Why adaptively sampling and weighting the feature is important for exploiting the within-scale characteristics and the cross-scale complementarity?

(2) Why using AFeB and AMS together could significantly improve the performance while only using one of them slightly improve the performance?

(3) In Table 7, only three baselines are investigated w.r.t. the model complexity, which is inadequate. More baselines should be investigated and compared.

**Limitations:**

yes

**Strengths And Weaknesses:**

Strengths

(1) Simultaneously taking the within-scale characteristics and the cross-scale complementarity into multi-scale architecture design increases the performance of single image denoising task. The reviewer agrees that the within-scale characteristics is significative for multi-scale architecture and single image denoising.

(2) The motivations of the proposed designs, i.e., MSANet and the three neural blocks, are clear and convincing. The experimental results show the effectiveness and robustness of the proposed designs.

(3) The authors conduct adequate experiments on both synthetic and real noise image datasets, and investigate the effectiveness of the proposed three blocks as well as the model complexity. Overall, the experiments are complete.

Weaknesses

(1) The proposed MSANet is not competitive in parameter numbers as there are different subnetworks corresponding to different scale features. This limits the use of the model on memory-constrained devices. In contrast, the running time and FLOPs are competitive due to multi-resolution features.

(2) Using AFeB and AMS together could significantly improve the performance, while using either AFeB or AMB alone slightly improve the performance, i.e., they are bound together for using. This limits the separate use of the two blocks.

---

> ### Author Response · Authors · 2022-08-02
> **Response to Reviewer 9gQL**
>
> **Q1: The proposed MSANet is not competitive in parameter numbers as there are different subnetworks corresponding to different scale features. This limits the use of the model on memory-constrained devices.**
>
> **A1:** Like other multi-scale networks, MSANet is less attractive in parameters, but its FLOPs are obviously lower than most baselines even with more parameters. Moreover, we would remind that one should pay more attention to the novelty and insight of this work to the community, i.e., the **within-scale characteristics (WSC)** of features are varying instead of fixed with the scales, which is our motivation for multi-scale architecture design and such a property is not reported so far as we known.
>
> **Q2: Using AFeB and AMS together could significantly improve the performance, while using either AFeB or AMB alone slightly improve the performance, i.e., they are bound together for using. This limits the separate use of the two blocks.**
>
> **Q4: Why using AFeB and AMS together could significantly improve the performance while only using one of them slightly improve the performance?**
>
> **A2 and A4:** AFeB and AMB together exploit the WSC. To exploit one WSC of high-resolution features, i.e., the mixture of details and noises, AFeB is designed for adaptively preserving the indispensable details and filtering unpleasant noises. Meanwhile, to improve another WSC of high-resolution features, i.e., the limited contextual information, AMB is used to enrich the contextual information and provide contextually informative features. For the low-resolution features, although contextual information is rich, it will destroy the image contents due to the limited resolution. To solve this problem, AMB is designed to enrich the contextual information while keeping the resolution unchanged. Therefore, suboptimal results will be obtained if using either AFeB or AMB alone, i.e., the WSC of multi-scale features is partially neglected.
>
> **Q3: Both AFeB and AFuB are based on deformable convolution to adaptively sample and weight the features, but one is used to adaptively select details and ﬁlter noises, the other is used to fuse the multi-scale features with varying characteristics. What caused this crucial difference? Why adaptively sampling and weighting the feature is important for exploiting the within-scale characteristics and the cross-scale complementarity?**
>
> **A3: First**, the roles of them played in network, and the information they need to achieve their missions. In brief, AFeB aims to exploit the WSC by preserving the image details and filtering unpleasant noises from their mixture. Therefore, AFeB needs the information from the features to distinguish the details and noises. AFuB aims to exploit the **cross-scale complementarity (CSC)** of multi-scale features with varying characteristics by transferring the fine-grained image details into the coarse-grained image contexts. Thus, AFuB simultaneously needs the contexts and details information to match the contexts with the corresponding details. **Second**, adaptively sampling and weighting the features could endow the modules with the capability of learning the sampling locations from features to indicate where are important for recovery, while assigning different weights based on features to show how important the locations are. As a result, AFeB could preserve the image details and filter unpleasant noises, and AFuB could transfer the fine-grained image details into the coarse-grained image contexts for better recovery performance.
>
> **Q5: More baselines should be investigated w.r.t. the model complexity in Table 7.**
>
> **A5:** Due to the time limitation, we will compare with more baselines in the next version.

---

> ### Author Response · Authors · 2022-08-06
> **To Reviewer 9gQL**
>
> Dear reviewer 9gQL,
>
> Thanks a lot for reviewing our paper and giving us the valuable suggestions.
>
> We have tried our best to answer all the questions according to the comments. We sincerely hope that our responses could address all your concerns. Is there anything that needs us to further clarify for the given concerns?
>
> Thanks again for your hard work.

---

### Official Review · Reviewer_rLGu · 2022-07-11

**Rating:** 7
**Confidence:** 5
**Soundness:** 2 fair
**Presentation:** 3 good
**Contribution:** 3 good

**Summary:**

This paper proposes Multi-Scale Adaptive Network (MSANet) for single image denoising with three blocks: Adaptive Feature Block (AFeB), Adaptive Multi-scale Block (AMB) and Adaptive Fusion Block (AFuB). The network architecture design explores the within-scale and cross-scale characteristics of multi-scale networks. Extensive experiments demonstrate that these presented modules gain better performance than the baseline network and some other denoising methods on several real or synthetic datasets.


**Questions:**

1) In Fig.1, the authors claim that the robustness of high-level and low-level scales is different. What is the definition of robustness there? Is there any experiment or reference to support this statement?

2) How does the motivation of exploring within-scale characteristics cause the design of the network architecture? The proposed three modules seem to be only simple combinations of residual blocks and deformable/dilated convolution. The performance improvements may come from the additional parameters rather than the designed architecture.

3) How do we combine the features weighted by channel attention and the features weighted by spatial attention in AMB？ The detailed process seems to be omitted.


**Strengths And Weaknesses:**

Strengths:

+This paper is well-written, and the proposed architecture is straightforward to follow.

+The summary of the difference between low-resolution and high-resolution scales in Fig.1 is valuable, and the motivation to explore the within-scale characteristics in architecture design is reasonable.

Weaknesses:

-The relationship between the motivation of exploring the within-scale characteristics and the design of the three proposed modules is farfetched. There is still a lack of related experiments or visualization results that prove that the presented modules can utilize the within-scale characteristics well.

-Comparison with the latest SOTAs such as MPRNet [1], Uformer [2], and DGUNet [3] is missing. The performances reported in this paper seem to be much worse than these previous denoising methods.

-The ablation study in Section 4.4 is limited to demonstrating the different characteristics of each module. Adding the proposed modules seems only to gain some improvements over the baseline network, but the in-depth analysis is missing. I can not get any intuitive conclusions from this part.

[1] Zamir, Syed Waqas, Aditya Arora, Salman Khan, Munawar Hayat, Fahad Shahbaz Khan, Ming-Hsuan Yang, and Ling Shao. Multi-stage progressive image restoration. In Proceedings of the IEEE/CVF Conference on Computer Vision and Pattern Recognition (CVPR) (2021)

[2] Wang, Zhendong, Xiaodong Cun, Jianmin Bao, Wengang Zhou, Jianzhuang Liu, and Houqiang Li. Uformer: A general u-shaped transformer for image restoration. In Proceedings of the IEEE/CVF Conference on Computer Vision and Pattern Recognition (CVPR) (2022)

[3] Mou, Chong, Qian Wang, and Jian Zhang. Deep Generalized Unfolding Networks for Image Restoration. In Proceedings of the IEEE/CVF Conference on Computer Vision and Pattern Recognition (CVPR) (2022)

---

> ### Author Response · Authors · 2022-08-02
> **Response to Reviewer rLGu (Part 1)**
>
> **Q1.1: The relationship between the motivation of exploring the within-scale characteristics and the design of the three proposed modules is farfetched.**
>
> **Q5.1: How does the motivation of exploring within-scale characteristics cause the design of the network architecture? The proposed three modules seem to be only simple combinations of residual blocks and deformable/dilated convolution.**
>
> **A1.1 and A5.1:** We would clarify that AFeB and AMB blocks are designed to exploit the **within-scale characteristics (WSC)**, and AFuB is designed to exploit the **cross-scale complementarity (CSC)**. In other words, not all these three blocks are developed for WSC. To exploit one WSC of high-resolution features, i.e., the mixture of details and noises, AFeB is designed for adaptively preserving the indispensable details and filtering unpleasant noises. Meanwhile, to improve another WSC of high-resolution features, i.e., the limited contextual information, AMB is used to enrich the contextual information and provide contextually informative features. For the low-resolution features, although contextual information is rich, it will destroy the image contents due to the limited resolution. To solve this problem, AMB is designed to enrich the contextual information while keeping the resolution unchanged.
>
> Hence, one could see that the proposed three modules are not SIMPLE combinations of residual blocks and deformable/dilated convolution. Instead, they are designed to achieve and implement our idea, i.e., different scale features show varying characteristics and should be processed by scale-speciﬁc structures rather than homologous architectures. Clearly, such an architecture design fashion is highly expected thanks to such a highly interpretable criterion.
>
> **Q1.2: There is still a lack of related experiments or visualization results that prove that the presented modules can utilize the within-scale characteristics well.**
>
> **A1.2:** We assume that the ablation studies (i.e., “AFeB+AMB”) and qualitative results could help to address this concern. In addition, in the new Supplementary Material, we add the qualitative results of intermediate features before and after all MSANet subnetworks at different scales to demonstrate that our modules can utilize the within-scale characteristics well.
>
> **Q2: Comparison with the latest SOTAs such as MPRNet, Uformer, and DGUNet is missing. The performances reported in this paper seem to be much worse than these previous denoising methods.**
>
> **A2:** We would highlight that one major contribution of this work is provide a novel insight to the community, i.e., different scale features show varying characteristics and should be processed by scale-specific structures rather than homologous architectures. This insight is model-agnostic which has the potential to design more powerful networks including Transformer and beyond. Due to limited time and resources, we have to conduct this work in the future. For the selection of baselines, we use the baselines that are comparable to our methods in parameters and FLOPs. Taking an image of 256$\times$256 as example, the parameters/FLOPs of MPRNet, Uformer, DGUNet respectively are 16M/1148G, 51M/179G, 17M/1729G, which are far more than our MSANet which is 8M/71G. Therefore, it is reasonable that they achieve better performance. In short and once again, we believe that the novel insights are more valuable (at least, equally important) to the community, and we will add related discussions in the next version.
>
> **Q3: The ablation study in Section 4.4 is limited to demonstrating the different characteristics of each module. Adding the proposed modules seems only to gain some improvements over the baseline network, but the in-depth analysis is missing. I cannot get any intuitive conclusions from this part.**
>
> **A3:** The ablation study is conducted to demonstrate the effectiveness of utilizing WSC and CSC. Specifically, “ED” and “ResB” are with homologous architectures, which use Identity Mapping and Residual Block to build subnetworks for different scale features, respectively. As mentioned in **A1**, AFeB and AMB together are used to exploit WSC, and AFuB is used to achieve CSC. As using AFeB and AMB alone cannot exploit WSC well, “AFeB”/“AMB” and “AFeB+AFuB”/“AMB+AFuB” only slightly improve the performance over “ResB” and “AFuB”, respectively. When using AFeB and AMB together, “AFeB+AMB” and MSANet (i.e., “AFeB+AMB+AFuB”) significantly improve the performance over “ResB” and “AFuB”, verifying our claim on the role of AFeB+AMB w.r.t. WSC. Furthermore, thanks to CSC from AFuB, “AFuB” and MSANet (“AFeB+AMB+AFuB”) are significantly better than “ResB” and “AFeB+AMB”, respectively. Therefore, the ablation study not only demonstrates the significance of well utilizing WSC and CSC, but also shows the effectiveness of the proposed solution.

---

> > ### Author Response · Authors · 2022-08-02
> > **Response to Reviewer rLGu (Part 2)**
> >
> > **Q4: In Fig.1, the authors claim that the robustness of high-level and low-level scales is different. What is the definition of robustness there? Is there any experiment or reference to support this statement?**
> >
> > **A4:** It refers to the robustness against noises. Specifically, compared with the high-resolution features, the low-resolution features contain less noises from the noisy input. To support this statement, we qualitatively show the features of different scales and visually illustrate them as Fig.1 in the new Supplementary Material.
> >
> > **Q5.2: The performance improvements may come from the additional parameters rather than the designed architecture.**
> >
> > **A5.2:** We would remind that, on the one hand, our ablation study has well investigated the effects of the proposed modules and the designed architecture with comparable parameters. On the other hand, some baselines such as CLEARER, RNAN take comparable even more parameters while their performance is obviously worse than our method. Besides, the FLOPs of our method are obviously lower than most baselines even with more parameters.
> >
> > **Q6: How do we combine the features weighted by channel attention and the features weighted by spatial attention in AMB?**
> >
> > **A6:** We perform the channel attention at first, and then perform the spatial attention.

---

> ### Author Response · Authors · 2022-08-06
> **To Reviewer rLGu**
>
> Dear reviewer rLGu,
>
> Thanks a lot for reviewing our paper and giving us the valuable suggestions.
>
> We have tried our best to answer all the questions according to the comments. We sincerely hope that our responses could address all your concerns. Is there anything that needs us to further clarify for the given concerns?
>
> Thanks again for your hard work.

---

> > ### Comment · Reviewer_rLGu · 2022-08-09
> > **Post-rebuttal**
> >
> > The authors have addressed all my concerns. I increase my score to Accept.

---

> > > ### Author Response · Authors · 2022-08-09
> > > **Response to Post-rebuttal**
> > >
> > > Thanks for your positive comments and suggestions. We would accordingly revise the problems and include some discussions about the concerns for a clearer presentation in the next version.

---

### Meta-Review · Area_Chair_K8xb · 2022-08-25

**Recommendation:** Accept
**Confidence:** Certain

**Metareview:**

All reviewers are positive about this paper. Although this paper does not achieve the best performance,  it reveals some  insights about some insights about scale characteristics of features, which is  model-agnostic and  potential to design more powerful networks. Also, the proposed method can reduce FlOPs obviously.

**Award:**

No

---

### Decision · Program_Chairs · 2022-09-14

Accept